**Data Availability Statement:** Sharing of individual participant data with third parties was not specifically included in the informed consent of the

# Eculizumab in patients with severe coronavirus disease 2019 (COVID-19) requiring continuous positive airway pressure ventilator support: *Retrospective cohort study*

Piero Ruggenenti[1,2], Fabiano Di Marco[3,4], Monica Cortinovis[1], Luca Lorini[5], Silvia Sala[5], Luca Novelli[3], Federico Raimondi[3,4], Sara Gastoldi[1], Miriam Galbusera[1], Roberta Donadelli[1], Caterina Mele[1], Rossella Piras[1], Marina Noris[1], Valentina Portalupi[2], Laura Cappelletti[2], Camillo Carrara[2], Federica Tomatis[2,6], Silvia Bernardi[2,6], Annalisa Perna[1], Tobia Peracchi[1], Olimpia Diadei[1], Ariela Benigni[1], Giuseppe Remuzzi[1] *

1 Istituto di Ricerche Farmacologiche Mario Negri IRCCS, Bergamo, Italy, 2 Unit of Nephrology and Dialysis, Azienda Socio-Sanitaria Territoriale (ASST) Papa Giovanni XXIII, Bergamo, Italy, 3 Unit of Pulmonary Medicine, Azienda Socio-Sanitaria Territoriale (ASST) Papa Giovanni XXIII, Bergamo, Italy, 4 Department of Health Sciences, University of Milan, Milan, Italy, 5 Intensive Care Unit, Azienda Socio-Sanitaria Territoriale (ASST) Papa Giovanni XXIII, Bergamo, Italy, 6 School of Nephrology, Università degli Studi di Milano, Milan, Italy

* giuseppe.remuzzi@marionegri.it

## Abstract

### Background

Complement activation contributes to lung dysfunction in coronavirus disease 2019 (COVID-19). We assessed whether C5 blockade with eculizumab could improve disease outcome.

### Methods

In this single-centre, academic, unblinded study two 900 mg eculizumab doses were added-on standard therapy in ten COVID-19 patients admitted from February 2020 to April 2020 and receiving Continuous-Positive-Airway-Pressure (CPAP) ventilator support from ≤24 hours. We compared their outcomes with those of 65 contemporary similar controls. Primary outcome was respiratory rate at one week of ventilator support. Secondary outcomes included the combined endpoint of mortality and discharge with chronic complications.

### Results

Baseline characteristics of eculizumab-treated patients and controls were similar. At baseline, sC5b-9 levels, *ex vivo* C5b-9 and thrombi deposition were increased. *Ex vivo* tests normalised in eculizumab-treated patients, but not in controls. In eculizumab-treated patients respiratory rate decreased from 26.8±7.3 breaths/min at baseline to 20.3±3.8 and 18.0±4.8 breaths/min at one and two weeks, respectively (p<0.05 for both), but did not change in controls. Between-group changes differed significantly at both time-points (p<0.01). Changes in

study, and unrestricted diffusion of such data may pose a potential threat of revealing participants' identities, as permanent data anonymisation was not carried out (patient records were instead de-identified per protocol during the data retention process). To minimise this risk, researchers who wish to inquire about access to individual participant data that underlie the results reported in this article shall submit a proposal to the Laboratory of Biostatistics of the Department of Renal Medicine of the Istituto di Ricerche Farmacologiche Mario Negri IRCCS (RenMedBiostatistics@marionegri.it). To gain access, data requestors will need to sign a data access agreement and obtain the approval of the local ethics committee.

**Funding:** The ASST Papa Giovanni XXIII in Bergamo (Italy) sponsored the trial, Brembo SpA (Curno, Bergamo, Italy) partially covered study costs by a liberal grant under the initiative "Progetto TrexUno" and Alexion Pharma Italy S.R.L. (Milan, Italy) freely supplied the study drug. Neither the sponsor nor the companies had any role in study design; in the collection, analysis and interpretation of data; in the writing of the report; and in the decision to submit the paper for publication. All authors had full access to all the data in the study and accept responsibility to submit for publication. There was no additional funding received for this study. The funder provided support in the form of salaries for authors PR, FDM, LL, SS, LN, VP, LC and CC, but did not have any additional role in the study design, data collection and analysis, decision to publish, or preparation of the manuscript. The specific roles of these authors are articulated in the 'author contributions' section.

**Competing interests:** M.G. reported grants from Omeros Corporation, Alexion Pharmaceuticals, F. Hoffman-La Roche Ltd and Novartis Pharma AG (payments were made to her institution), all outside the submitted work; M.N. reported grants from Omeros Corporation, Novartis Pharma AG, F. Hoffman-La Roche Ltd and BioCryst Pharmaceuticals (payments were made to her institution) as well as personal fees from Inception Sciences, BioCryst Pharmaceuticals and Alexion Pharmaceuticals, all outside the submitted work; A. B. reported personal fees from Akebia Pharmaceuticals, Alexion Pharmaceuticals, BioCryst Pharmaceuticals, Janssen Research & Development LLC, as well as speaker honorarium/ travel reimbursements from Alnylam, Boehringer Ingelheim and Inception Sciences Canada, all outside the submitted work; G.R. reported personal fees from Akebia Pharmaceuticals, Alexion Pharmaceuticals, BioCryst Pharmaceuticals and

respiratory rate correlated with concomitant changes in *ex vivo* C5b-9 deposits at one (rs = 0.706, p = 0.010) and two (rs = 0.751, p = 0.032) weeks. Over a median (IQR) period of 47.0 (14.0–121.0) days, four eculizumab-treated patients died or had chronic complications versus 52 controls [$HR_{Crude}$ (95% CI): 0.26 (0.09–0.72), p = 0.010]. Between-group difference was significant even after adjustment for age, sex and baseline serum creatinine [$HR_{Adjusted}$ (95% CI): 0.30 (0.10–0.84), p = 0.023]. Six patients and 13 controls were discharged without complications [$HR_{Crude}$ (95% CI): 2.88 (1.08–7.70), p = 0.035]. Eculizumab was tolerated well. The main study limitations were the relatively small sample size and the non-randomised design.

## Conclusions

In patients with severe COVID-19, eculizumab safely improved respiratory dysfunction and decreased the combined endpoint of mortality and discharge with chronic complications. Findings need confirmation in randomised controlled trials.

## Introduction

As observed in other experimental and human forms of coronavirus lung infection [1,2], dramatic respiratory dysfunction [3] and dismal outcomes [4] of Severe-Acute-Respiratory-Syndrome Coronavirus-2 (SARS-CoV-2) 2019, or COVID-19 [3,5], are largely mediated by overwhelming release of pro-inflammatory cytokines (cytokine storm) [6] and uncontrolled complement activation [7,8]. Endothelial injury and microangiopathic lesions similar to those observed in the hemolytic uremic syndrome (HUS) [9,10] and deposits of C5b-9 in lung and skin vessels [10], as well as in glomeruli and tubuli [11] of patients dying of COVID-19 confirm that complement activation, in particular of the terminal pathway, may have a key pathogenic role in COVID-19 [12].

Eculizumab is a humanised anti-C5 monoclonal antibody approved for the treatment of paroxysmal nocturnal hemoglobinuria and atypical HUS [13,14]. FDA approved a program of eculizumab off-label compassionate use for the treatment of non-intubated patients with COVID-19. Initial case series and explorative studies showed encouraging effects of C5 blockade in patients with COVID-19 [15–17], even in combination with the JAK1/2 inhibitor ruxolitinib [16]. Based on this background we planned to add two 900 mg eculizumab doses on best supportive therapy in ten patients with COVID-19 who required Continuous-Positive-Airway-Pressure (CPAP) ventilator support and compared their outcomes with outcomes of similar contemporary controls who received the same supportive therapy, but no eculizumab.

## Materials and methods

### Study population

We included adults admitted at the Azienda-Socio-Sanitaria-Territoriale (ASST) Papa Giovanni XXIII in Bergamo (Italy) because of severe COVID-19 who were receiving CPAP ventilator support from 24 hours or less. Diagnosis was based on WHO Interim guidance criteria [18], and confirmed by detection at admission of SARS-CoV-2 genome from nasal swabs and respiratory samples by using two different molecular methods (GeneFinder COVID-19-Elitech Group, Allplex™ 2019-nCoV Assay—Seegene Inc) according to the manufacturer's instructions and WHO protocol (Supplementary Methods in S1 Appendix) [18].

Janssen Research & Development LLC, as well as speaker honorarium/travel reimbursements from Alnylam, Boehringer Ingelheim and Inception Sciences Canada, all outside the submitted work. This does not alter our adherence to PLOS ONE policies on sharing data and materials. All the other authors have nothing to disclose.

**Abbreviations:** COVID-19, coronavirus disease 2019; CPAP, continuous positive airway pressure; CRP, C-reactive protein; $FiO_2$, fractional inspired oxygen; HMEC-1, human microvacular endothelial cells; HUS, hemolytic uremic syndrome; $PaCO_2$, arterial partial pressure of carbon dioxide; $PaO_2$, arterial partial pressure of oxygen, $PaO_2/FiO_2$, ratio of partial pressure of arterial oxygen to fractional inspired oxygen.

Ten consenting participants received compassionate eculizumab treatment along with anti-Neisseria Meningitis and Pneumococcus antibiotic coverage. The drug was freely supplied by the manufacturer (Alexion Pharma Italy S.R.L., Milan). No participant received compensation. Then, when all patients had completed eculizumab therapy, outcomes of eculizumab-treated patients and contemporary controls were compared retrospectively. The prospective compassionate treatment protocol and the retrospective controlled study were both approved by the Ethical Committee of Bergamo. The Ethical Committee that approved our retrospective observational study stated that the informed consent to participate in the study and to use medical records for research purposes had to be collected whenever the patient could be contacted. Thus, we collected the written informed consent from those eculizumab-treated patients and controls who were still alive at the time the study was conducted.

## Eculizumab compassionate treatment

Nine-hundreds mg of eculizumab were intravenously infused within 24 hours of CPAP ventilator support and seven to ten days thereafter (this flexibility was justified by the emergency context). The number of patients given eculizumab therapy depended on drug availability, while the selection of eculizumab recipients was at the discretion of the treating physician and influenced by logistic reasons. Indeed, to avoid overlaps of drug administrations and specific laboratory tests to monitor treatment effects in different patients, and prevent possible interference with clinical patient management, we administered eculizumab to one patient every six to seven consecutive potential candidates (Supplementary Methods in S1 Appendix). In no case patients eligible to receive eculizumab objected to the use of their medical data for research purposes. Nonetheless, patients would still have been eligible to receive eculizumab if they had objected to the use of their data in medical research. No systematic change in supportive treatment was introduced. Patients involved in ongoing clinical trials were excluded.

## Follow-up

All participants were followed up to death or hospital discharge. History, vital signs, clinical, laboratory and safety parameters and adverse events were recorded in patients' medical records.

Blood samples to evaluate a marker of complement activation in plasma (sC5b-9) and *ex vivo* serum-induced complement deposition and thrombus formation on ADP-activated human microvascular endothelial cells (HMEC-1) of dermal origin [19–21] were collected from all eculizumab-treated patients (before eculizumab administration) and four controls ("Biochemical Controls") within 24 hours from CPAP initiation (baseline) along with samples for genetic analyses. *In vivo* and *ex vivo* tests were repeated 1–4 days and 7–16 days after baseline, respectively, and 30–60 days after discharge (recovery visit). Because of resource restrictions, *ex vivo* thrombi deposition at 7–16 days was not evaluated. Complement and genetic analyses were freely performed at the Laboratories of the Istituto di Ricerche Farmacologiche Mario Negri IRCCS in Bergamo (Italy). Other laboratory parameters were evaluated at the hospital clinical laboratories. All data were recorded in the same dedicated database (further details in Study Protocol in S2 Appendix).

## Complement evaluation and genotyping

SC5b-9 levels were evaluated in plasma EDTA by MicroVue SC5b-9 Plus EIA (SC5b-9 Plus; Quidel). *Ex-vivo* serum-induced C5b-9 deposits and thrombus formation on HMEC-1 lines were evaluated as described previously, with minor modifications (Supplementary Methods in S1 Appendix and S1 Fig) [19–21]. CFH, MCP, CFI, CFB, C3, THBD coding sequences were

screened by amplicon-based next generation sequencing and H3 CFH haplotype and CFH/CFHR genomic abnormalities were evaluated as previously described (Supplementary Methods in S1 Appendix) [22].

## Sample size and statistical analyses

We calculated that with the administration of two 900 mg doses per patient, drug supply would have been sufficient to treat ten patients. Considering that eculizumab would have been administered to one patient every six to seven potential candidates we predicted that at inclusion of ten eculizumab-treated patients we would have identified a total of 60 to 70 potential controls. Among potential controls we selected those with the same age-range of eculizumab-treated patients.

Due to the retrospective and observational nature of the study the sample size was not calculated. Primary outcome was the absolute change in respiratory rate at week one versus baseline. Secondary short-term outcomes included changes in respiratory rate at two weeks and changes in arterial partial pressure of oxygen ($PaO_2$) and carbon dioxide ($PaCO_2$), $PaO_2$ to fractional inspired oxygen ($FiO_2$) ratio ($PaO_2/FiO_2$); heart rate, systolic, diastolic and mean blood pressure; C-Reactive Protein (CRP), D-dimer test, serum creatinine, blood cell count at one and two weeks after baseline and concomitant changes in complement parameters in a subgroup. Complement parameters were evaluated also at recovery visits.

The combined endpoint of in-hospital death or discharge with chronic severe complications was considered as the most clinically relevant long-term outcome. Secondary long-term outcomes included death during hospitalisation as single endpoint and discharge without chronic complications. Any serious adverse event was recorded and monitored up to patient discharge or death.

We analysed continuous variables through descriptive statistics. We reported data as mean (SD) or median [IQR], within-group changes vs. baseline with paired t-test or Wilcoxon signed-rank test, and between-group differences by analysis of covariance (ANCOVA) or chi-square or Fisher's exact test, as appropriate. Survival analysis was performed by means of Cox proportional hazard regression models and results were expressed as hazard ratio (HR) and 95% confidence interval (CI). Analyses were adjusted by pre-defined covariates age and sex, and serum creatinine, the only covariate that, among baseline covariates listed in Table 1, at explorative univariable analysis significantly associated with all considered long-term out-

**Table 1. Patient characteristics at baseline according to treatment group.**

| | Overall (n = 75) | Eculizumab (n = 10) | Controls (n = 65) |
|---|---|---|---|
| **Demographic and clinical characteristics** | | | |
| Age, *years* | 65.5 ± 12.9 | 60.0 ± 15.1 | 66.1 ± 12.5 |
| Males, *n (%)* | 56 (75) | 7 (70) | 49 (75) |
| Smokers, *n (%)* | 12 (16) | 2 (20) | 10 (15) |
| Comorbidities per patient | 2 [1–3] | 1 [0–2] | 2 [1–3] |
| Patients without comorbidities, *n (%)* | 24 (32) | 3 (30) | 21 (32) |
| Hypertension alone, *n (%)* | 10 (13) | 1 (10) | 9 (14) |
| Other comorbidities and hypertension, *n (%)* | 27 (36) | 1 (10) | 26 (40) |
| Other comorbidities without hypertension, *n (%)* | 14 (19) | 5 (50) | 9 (14) |
| Systolic blood pressure, *mmHg* | 131.7 ± 23.3 | 139.7 ± 37.6 | 130.4 ± 20.2 |
| Diastolic blood pressure, *mmHg* | 73.7 ± 12.0 | 77.6 ± 7.0 | 73.1 ± 12.6 |
| MAP, *mmHg* | 93.1 ± 13.7 | 98.3 ± 13.5 | 92.2 ± 13.7 |

(*Continued*)

**Table 1.** (Continued)

|  | Overall (n = 75) | Eculizumab (n = 10) | Controls (n = 65) |
|---|---|---|---|
| Heart rate, *bpm* | 82.9 ± 14.0 | 85.4 ± 12.5 | 82.5 ±14.3 |
| **Respiratory functional parameters** |  |  |  |
| Respiratory rate, *breaths/min* | 26.6 ± 7.4 | 26.8 ± 7.3 | 26.6 ± 7.5 |
| $PaO_2$, *mmHg* | 70.5 [59.6–92.0] | 80.5 [66.0–90.0] | 69.0 [58.0–92.0] |
| $PaCO_2$, *mmHg* | 32.0 [30.0–37.0] | 31.5 [30.0–37.0] | 32.0 [29.8–36.5] |
| $PaO_2/FiO_2$, *mmHg* | 125.7 [100.0–183.8] | 138.1 [110.0–159.3] | 124.3 [97.1–184.0] |
| $FiO_2$ | 0.6 [0.5–0.7] | 0.6 [0.6–0.6] | 0.6 [0.5–0.7] |
| Arterial pH | 7.46 [7.44–7.49] | 7.47 [7.44–7.48] | 7.46 [7.44–7.50] |
| **Laboratory parameters** |  |  |  |
| White blood cell count, $x10^9/L$ | 8.12 [5.89–12.41] | 10.72 [7.54–12.41] | 7.94 [5.75–12.26] |
| Neutrophil count, $x10^9/L$ | 6.10 [4.39–9.58] | 8.79 [6.02–11.31] | 5.85 [4.08–8.95] |
| Lymphocyte count, $x10^9/L$ | 0.75 [0.52–1.12] | 0.63 [0.44–0.76] | 0.79 [0.57–1.14] |
| Monocyte count, $x10^9/L$ | 0.38 [0.28–0.51] | 0.42 [0.32–0.49] | 0.37 [0.28–0.55] |
| Platelet count, $x10^9/L$ | 237.0 [150.0–308.0] | 303.5 [230.0–447.0][§] | 229.5 [137.0–286.5] |
| NLR | 7.44 [4.83–19.37] | 16.88 [7.67–20.08] | 6.91 [4.68–12.63] |
| PLR | 264.3 [187.7–514.4] | 402.3 [280.3–879.3] [§] | 244.4 [176.2–500.0] |
| Hemoglobin, *g/dL* | 13.4 [12.1–14.7] | 13.7 [12.2–14.7] | 13.4 [12.0–14.6] |
| C-reactive protein, *mg/dL* | 14.4 [9.1–21.9] | 15.4 [11.2–22.1] | 14.2 [9.1–21.5] |
| Aspartate aminotransferase, *U/L* | 53.0 [36.0–81.5] | 42.0 [35.5–50.5] | 56.0 [36.5–84.5] |
| Alanine aminotransferase, *U/L* | 44.0 [31.0–76.0] | 43.5 [29.0–69.0] | 44.0 [31.0–76.0] |
| LDH, *U/L* | 610 ± 301 | 499 ± 144 | 626 ± 315 |
| Serum creatinine, *mg/dL* | 0.89 [0.70–1.09] | 0.68 [0.60–0.95] | 0.90 [0.73–1.09] |
| Estimated GFR, *mL/min*[#] | 67.0 ± 23.2 | 79.7 ± 25.0 | 65.1 ±22.5 |
| D-dimer, *ng/mL*[*] | 942 [624–2132] | 871 [568–1044] | 1002 [624–2360] |
| IL-6, *pg/mL*[°] | 86.5 [53.2–132.0] | 83.6 [25.3–121.9] | 87.3 [53.2–132.0] |
| sC5b-9, *ng/mL*[‡] | 1022.0 ± 461.2 | 1145.1 ± 458.0 | 612.5 ± 72.5 |
| C5b-9, *% increase versus control*[¥] | 276.6 ± 71.7 | 270.3 ± 69.4 | 292.3 ± 85.7 |
| Thrombus formation, *pixel*[†] | 2821.8 ± 1326.7 | 2881.9 ± 1434.7 | 2661.7 ± 1242.5 |
| **Patients with medications, *n (%)*** |  |  |  |
| ACE inhibitors or ARB[&] | 26 (35) | 0 (0) [§] | 26 (40) |
| Darunavir/cobicistat | 39 (52) | 7 (70) | 32 (49) |
| Hydroxychloroquine | 51 (68) | 9 (90) | 42 (65) |

Data are numbers (percentages), mean ± SD or median [IQR], as appropriate. Abbreviations: ACE, angiotensin-converting enzyme; ARB, angiotensin II receptor blockers; $FiO_2$, fraction of inspired oxygen; GFR, glomerular filtration rate; IL-6, interleukin-6; LDH, lactate dehydrogenase; MAP, mean arterial pressure; NLR, neutrophil-to-lymphocyte ratio; $PaCO_2$, partial pressure of arterial carbon dioxide; $PaO_2/FiO_2$, ratio of partial pressure of arterial oxygen to fractional inspired oxygen; PLR, platelet-to-lymphocyte ratio. Comorbidities: Hypertension, cancer, cardiovascular disease, cerebrovascular disease, chronic kidney disease, chronic liver disease, chronic obstructive pulmonary disease, diabetes mellitus and obesity.

[#]Estimated through the Chronic Kidney Disease Epidemiology Collaboration (CKD-EPI) equation.

[*]D-dimer measurement was available in 8 patients of the eculizumab group and in 48 patients of the control group.

[°]IL-6 measurement was available in 8 patients of the eculizumab group and in 42 patients of the control group.

[‡] sC5b-9 levels were available in 10 patients of the eculizumab group and in 3 patients in the control group.

[¥] C5b-9 was available in 10 patients of the eculizumab group and in 4 patients in the control group.

[†]Thrombus formation was available in 8 patients of the eculizumab group and in 3 patients in the control group.

[&]In the control group 17 patients were given an ACE inhibitor and nine an ARB.

[§]P<0.05 versus controls.

comes. Cumulative events were constructed using the Kaplan-Meier method. Correlation analysis was carried out using Pearson's r or Spearman's rho correlation coefficient [23]. A further exploratory repeated measures correlation analysis was considered using package 'rmcorr'. No imputation method was used for missing values. Data were analysed by SAS (version 9.4), STATA (version 15) and package 'rmcorr' (version 0.4.1). Statistically significant differences were assumed at 5% level of probability.

## Results

From February 2020 to April 2020 we included 75 participants: ten eculizumab-treated patients and 65 controls. All of them received CPAP ventilator support from 24 hours or less and standard treatment including low-dose steroid, low-molecular-weight heparin and anti-microbial prophylaxis with cephtriaxone and azithromycin (which served also as prophylaxis for Neisseria Meningitis and Pneumococcus infection in eculizumab-treated patients). Most patients received also hydroxycholoroquine and/or antiviral therapy (darunavir and cobicistat combination). Tachypnea, hypoxia, hypocapnia, markedly decreased $PaO_2/FiO_2$ ratio and increased $FiO_2$ were consistent with severe respiratory distress at inclusion (Table 1). Other detailed demographic, clinical and laboratory parameters are reported in Table 1 and Supplementary Methods (S1 Appendix).

### Complement activity and ex vivo complement deposition and thrombogenesis

At baseline, sC5b-9 plasma levels were significantly higher in COVID-19 patients than in 10 contemporary healthy controls and even as compared to levels observed in historical patients with atypical HUS (Fig 1A). Sera from all the 14 analysed patients induced strong C5b-9 deposition (>149% of a control serum pool) on ADP-activated HMEC-1 (Fig 1B). Pre-exposure to sera from 11 patients induced massive thrombus formation on ADP-activated HMEC-1 flowed with normal heparinised whole blood added with mepacrine (S1 Fig). C5b-9 and thrombus deposition were significantly stronger than those induced by sera from healthy controls and similar to those induced by sera from historical patients with atypical HUS (Fig 1B and 1C).

### Genotyping

Genetic analysis was done in all eculizumab-treated COVID-19 patients and four COVID-19 controls ("Biochemical Controls"). Next generation sequencing screening revealed only a new missense heterozygous variant in C3 (c.C1426A, p.L476I) in one of the four COVID-19 controls. This variant has not been reported in patients with atypical HUS or other complement-related genetic diseases. No functional studies are available and the variant is predicted damaging only by two of 12 in silico tools (CADD 9.844). We classified the p.L476I as variant of unknown significance. The H3 CFH haplotype, which had been previously associated with the risk of atypical HUS [24] and with reduced plasma levels of factor H, was identified only in 2 patients (both are heterozygous) with an allele frequency of 0.11, which is not different from the allele frequency (0.17) that we previously reported in healthy subjects [25].

Copy number variation analysis in the genomic region including CFH and the 5 CFHR genes revealed the common CFHR3-CFHR1 heterozygous deletion in 6 out of 14 patients (frequency of the deleted allele 0.21 compared to 0.19 in 100 healthy controls, p = 0.6 Fisher's exact test). We did not identify any CFH/CFHR hybrid gene or other rare genomic rearrangements.

**A**

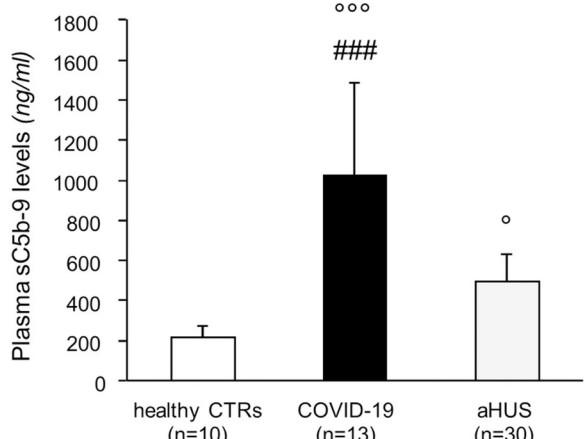

**B**

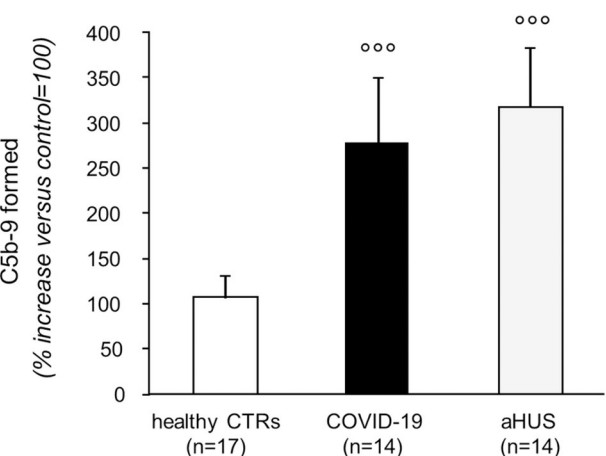

**C**

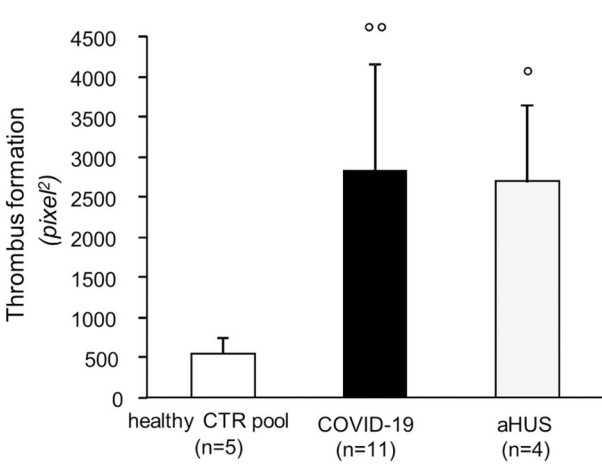

**Fig 1. Circulating complement profile and ex-vivo effect of serum on C5b-9 deposition and thrombus formation on HMEC-1 in COVID-19 patients.** (A) SC5b-9 plasma levels in ten healthy subjects (healthy CTRs, negative controls), in 13 COVID-19 patients evaluated within 24 hour initiation of CPAP ventilator support and in patients with atypical hemolytic uremic syndrome studied in remission (aHUS, positive controls, n = 30). (B) Effect of serum from COVID-19 patients on C5b-9 formation on microvascular endothelial cells (HMEC-1). ADP-activated HMEC-1 were incubated for 2 hr with 50% serum, diluted with test medium (HBSS with 0.5% BSA), from healthy subjects (healthy CTRs, n = 17), or COVID-19 patients evaluated within 24 hour initiation of CPAP ventilator support (n = 14), or aHUS patients studied in remission (aHUS positive controls, n = 14) or with a pool of sera from healthy controls. At the end of incubation, cells were washed, fixed, and stained with rabbit anti-human complement C5b-9 complex antibody followed by FITC-conjugated secondary antibody. An AXIO Image.Z2 laser microscope was used to view the fluorescent staining on the endothelial cell surface, the HMEC-1 area covered by C5b-9 staining was calculated by automatic edge detection (Image J software), and values were expressed as the percentage of C5b-9 deposits induced by a pool of sera run in parallel (control, reference 100%). (C) Effect of serum from COVID-19 patients on thrombus formation on microvascular endothelial cells (HMEC-1). HMEC-1 were treated with ADP and exposed for 2 h in static conditions to 50% serum, diluted with test medium, from COVID-19 patients evaluated within 24 hour initiation of CPAP ventilator support (n = 11), or aHUS patients studied in remission (aHUS positive controls, n = 4) or with a pool of sera from healthy controls (healthy CTR pool). Perfusion of heparinised whole blood from healthy subjects (added with mepacrine) was then performed in a thermostatic flow chamber in which one surface of the perfusion channel was a glass slide seeded with a monolayer of endothelial cells, at a constant flow rate of $1500 \, sec^{-1}$ (60 dynes/$cm^2$). After 3 min, perfusion was stopped, and the slide with the endothelial cell monolayer was dehydrated and fixed. A confocal inverted laser microscope was used to view the fluorescent staining on the endothelial cell surface, and the HMEC-1 area covered by thrombi was calculated by automatic edge detection (Image J software), and values expressed in $pixels^2$. Data are mean ± SD. One-way Anova test. Abbreviations: CTRs, controls °°°P<0.0001, °°P<0.01, °P<0.05 versus healthy CTRs. ###P<0.0001 versus aHUS.

## Eculizumab compassionate treatment

One patient received only the first eculizumab dose because he was transferred to another hospital before the second dose could be administered. His data were recorded and analysed as for the other patients. Baseline characteristics of eculizumab-treated patients were similar to those of the whole patient population (Table 1).

Eculizumab was well tolerated and no treatment-related adverse event was observed. Eculizumab was associated with a significant reduction in respiratory rate at one and two weeks that was paralleled by a concomitant increase in arterial $PaCO_2$ and a decrease in mean and diastolic blood pressure (Table 2). CRP and lactate dehydrogenase serum levels decreased as well, whereas estimated glomerular filtration rate significantly increased. Other changes are shown in Table 2.

Plasma sC5b-9 levels significantly decreased after the first eculizumab administration versus baseline, but mean values persistently exceeded normal range and fully normalised only at post-discharge recovery visits (Fig 2A). Serum-induced *ex-vivo* C5b-9 deposition and thrombus formation on HMEC-1 significantly and persistently decreased to normal range after eculizumab administration up to post-discharge recovery visits (Fig 2B and 2C).

## Controlled study

Baseline characteristics were similar between eculizumab-treated patients and controls (Table 1), with the exception of lower platelet count and platelet-to-lymphocyte ratio, and more frequent use of renin-angiotensin system (RAS) blockers in controls. In the four "biochemical" controls baseline sC5b-9 plasma levels, *ex-vivo* serum-induced C5b-9 deposition and thrombus formation were similarly elevated as in eculizumab-treated patients (Fig 2A–2C).

Respiratory rate did not change appreciably at one and two weeks of observation vs. baseline in controls, whereas arterial $PaCO_2$ significantly increased at both time-points. No significant changes were observed in heart rate and arterial blood pressure. Changes in other considered parameters are shown in Table 2. Changes in respiratory rate, $PaCO_2$, mean and

**Table 2. Clinical, respiratory and laboratory parameters at baseline and during follow-up according to study group.**

| | Eculizumab (n = 10) | | | Controls (n = 65) | | |
|---|---|---|---|---|---|---|
| | *Baseline* | *Week 1* | *Week 2* | *Baseline* | *Week 1* | *Week 2* |
| **Clinical parameters** | | | | | | |
| Systolic BP, *mmHg* | 139.7±37.6 | 129.2±13.1 | 117.0±11.1[#] | 130.4±20.2 | 131.6±19.8 | 128.6±21.5 |
| Diastolic BP, *mmHg* | 77.6±7.0 | 68.9±9.5[##] | 65.4±12.2[*####] | 73.1±12.6 | 72.4±13.1 | 71.3±11.9 |
| MAP, *mmHg* | 98.3±13.5 | 89.0±8.5[#] | 82.6±7.7[*####] | 92.2±13.7 | 92.1±13.6 | 90.4±13.3 |
| Heart rate, *bpm* | 85.4±12.5 | 79.1±8.2[#] | 81.2±9.8 | 82.5±14.3 | 82.9±14.8 | 80.9±12.7 |
| **Respiratory parameters** | | | | | | |
| Respiratory rate, *breaths/min* | 26.8±7.3 | 20.3±3.8[*##] | 18.0±4.8[*##] | 26.6±7.5 | 26.0±7.4 | 24.3±6.5 |
| PaO$_2$, *mmHg* | 80.5 [66.0–90.0] | 78.0 [70.0–89.0] | 75.1 [63.0–88.0] | 69.0 [58.0–92.0] | 69.0 [58.0–90.4] | 69.5 [57.0–89.0] |
| PaCO$_2$, *mmHg* | 31.5 [30.0–37.0] | 55.0 [36.5–58.9][** #] | 54.7 [38.6–68.4] | 32.0 [29.8–36.5] | 41.0 [34.0–47.0][***] | 40.3 [32.7–48.0][**] |
| PaO$_2$/FiO$_2$, *mmHg* | 138.1 [110.0–159.3] | 162.6 [100.0–180.0] | 97.6 [75.0–179.3] | 124.3 [97.1–184.0] | 95.7 [71.4–142.9] | 111.1 [90.0–162.9] |
| FiO$_2$ | 0.6 [0.6–0.6] | 0.6 [0.3–0.8] | 0.3 [0.2–0.8] | 0.6 [0.5–0.7] | 0.7 [0.5–0.8][*] | 0.6 [0.4–0.7] |
| Arterial pH | 7.47 [7.44–7.48] | 7.38 [7.38–7.39] | 7.41 [7.36–7.44] | 7.46 [7.44–7.50] | 7.44 [7.40–7.48] | 7.45 [7.41–7.48] |
| **Laboratory parameters** | | | | | | |
| WBC count, *x10$^9$/L* | 10.72 [7.54–12.41] | 10.65 [8.84–12.80] | 8.23 [6.59–12.69] | 7.94 [5.75–12.26] | 12.15 [7.64–17.16][***] | 12.24 [9.96–15.46][***] |
| Neutrophil count, *x10$^9$/L* | 8.79 [6.02–11.31] | 8.91 [7.50–11.67] | 5.50 [4.19–10.12] | 5.85 [4.08–8.95] | 11.09 [6.84–14.89][***] | 10.01 [7.45–12.54][***] |
| Lymphocyte count, *x10$^9$/L* | 0.63 [0.44–0.76] | 0.76 [0.34–1.02] | 1.04 [0.89–1.99][*] | 0.79 [0.57–1.14] | 0.66 [0.45–1.08] | 1.32 [0.70–1.95][**] |
| Monocyte count, *x10$^9$/L* | 0.42 [0.32–0.49] | 0.62 [0.52–0.86][*] | 0.67 [0.53–0.73][**] | 0.37 [0.28–0.55] | 0.55 [0.32–0.76][***] | 0.80 [0.58–0.93][***] |
| Platelet count, *x10$^9$/L* | 303.5 [230.0–447.0][#] | 306.5 [251.0–378.0] | 192.0 [177.0–317.0] | 229.5 [137.0–286.5] | 244.0 [165.0–333.0] | 243.5 [184.0–335·5] |
| NLR | 16.88 [7.67–20.08] | 9.51 [8.44–15.46] | 5.09 [2.08–16.85] | 6.91 [4.68–12.63] | 15.74 [9.27–34.50] | 6.91 [3.36–14.64] |
| PLR | 402.3 [280.3–879.3][#] | 322.3 [255.4–561.3] | 157.4 [98.9–388.3] | 244.4 [176.2–500.0] | 367.7 [178.6–545.8] | 244.3 [86.3–383.5] |
| Hemoglobin, *g/dL* | 13.7 [12.2–14.7] | 13.3 [11.0–13.7] | 11.4 [10.0–13.0][*] | 13.4 [12.0–14.6] | 11.8 [10.9–13.1] [***] | 11.3 [9.7–12.4] [***] |
| C-reactive protein, *mg/dL* | 15.4 [11.2–22.1] | 1.0 [0.5–1.4][**##] | 0.8 [0.2–9.5][*] | 14.2 [9.1–21.5] | 5.9 [2.0–15.9][***] | 4.2 [0.7–8.4][***] |
| AST, *U/L* | 42..0 [35.5–50.5] | 32.5 [21.0–46.0] | 27.0 [20.5–46.0] | 56.0 [36.5–84.5] | 36.0 [26.0–48.0][***] | 27.0 [20.0–40.5][***] |
| ALT, *U/L* | 43.5 [29.0–69.0] | 86.5 [61.0–111.0] [**#] | 68.0 [52.0–96.5][*] | 44.0 [31.0–76.0] | 53.0 [34.0–77.0][**] | 59.0 [34.0–94.0][**] |
| LDH, *U/L* | 499±144 | 301±112[***] | 267±73[***] | 626±315 | 446±192[***] | 388±166[**] |
| Serum creatinine, *mg/dL* | 0.68 [0.60–0.95] | 0.56 [0.52–0.73] | 0.56 [0.49–0.74] | 0.90 [0.73–1.09] | 0.79 [0.61–1.00][*] | 0.70 [0.52–0.88][**] |
| Estimated GFR, *mL/min* | 79.7±25.0 | 90.3±22.1[**] | 89.9±28.9[*] | 65.1±22.5 | 71.8±22.8[***] | 76.8±22.8[***] |
| D-dimer, *ng/mL* | 871 [568–1044] | 916 [378–2551] | 1336 [278–3370] | 1002 [624–2360] | 3139 [1045–6306][***] | 2743 [1072–4008][***] |
| IL-6, *pg/mL* | 83.6 [25.3–121.9] | 8.0 [7.6–11.5] | 49.2 [38.9–84.3] | 87.3 [53.2–132.0] | 35.8 [2.0–58.5] | 19.9 [2.0–37.8] |
| sC5b-9, *ng/mL*[‡] | 1145.1±458.0 | 761.5±247.3[*] | 831.5±211.2 | 612.5±72.5 | 578.6±144.5 | 704.8±192.7 |
| C5b-9, *% increase vs control*[¥] | 270.3±69.4 | 78.5±19.0[***###] | 128.8±54.3[*###] | 292.3±85.7 | 316.5±37.6 | 322·4±44·1 |
| Thrombus formation, *pixel$^2$*† | 2881.9±1434.7 | 638.1±251.4[**###] | | 2661.7±1242.5 | 3593.7±1018.3 | |

Data are mean (SD) or median [IQR], as appropriate. Abbreviations: ALT, alanine aminotransferase; AST, aspartate aminotransferase; BP, blood pressure; CRP, C-reactive protein; GFR, glomerular filtration rate; IL-6, interleukin-6; LDH, lactate dehydrogenase; MAP, mean arterial pressure; NLR, neutrophil-to-lymphocyte ratio; PaCO$_2$, partial pressure of arterial carbon dioxide; PaO$_2$/FiO$_2$, ratio of partial pressure of arterial oxygen to fractional inspired oxygen; PLR, platelet-to-lymphocyte ratio.
[‡] sC5b-9 levels were available in 10 patients of the eculizumab group and in 3 patients in the control group.
[¥] C5b-9 was available in 10 patients of the eculizumab group and in 4 patients in the control group.
†Thrombus formation was available in 8 patients of the eculizumab group and in 3 patients in the control group. For complement activity, ex vivo complement deposition and thrombus formation, Week 1 corresponds to 1–4 days post-CPAP, and Week 2 corresponds to 7–16 days post-CPAP. T-test or Wilcoxon Signed Rank
*p<0.05, **p<0.01, ***p<0.001 vs baseline, ANCOVA test #p<0.05, ##p<0.01, ###p<0.001 vs controls.

diastolic blood pressure, and median CRP values at one and two weeks (Fig 3A–3D) versus baseline significantly differed between treatment groups (Table 2). Other changes are shown in Table 2.

*Ex-vivo* C5b-9 and thrombus deposition in "biochemical" controls did not change at one and two weeks vs. baseline and normalised only at recovery visits (Fig 2B and 2C). Thus,

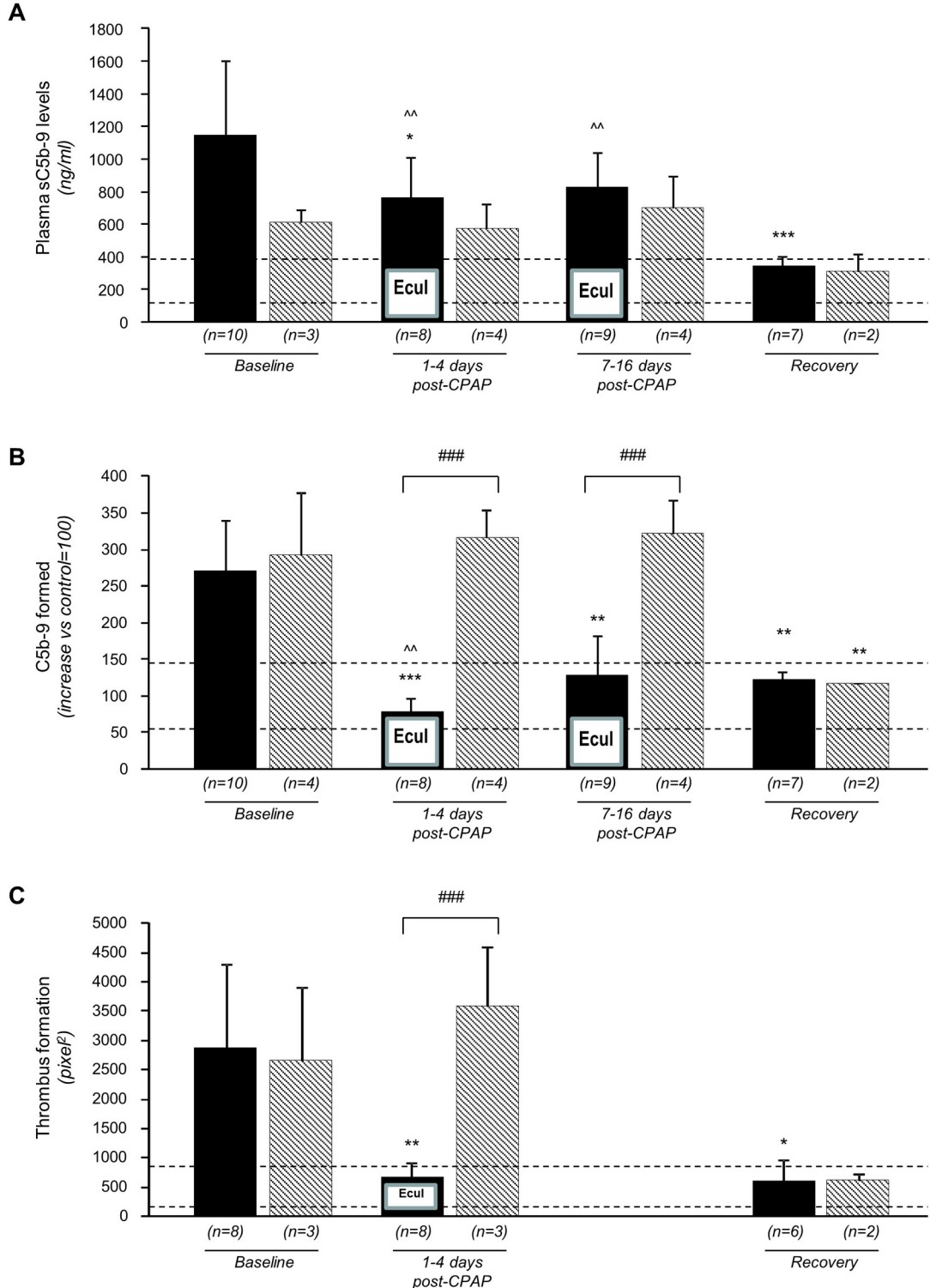

**Fig 2. Changes in sC5b-9 plasma levels and in ex vivo serum induced C5b-9 and thrombus formation on HMEC-1 lines in eculizumab-treated patients and "biochemical" controls.** SC5b-9 plasma levels (A), *ex vivo* serum induced C5b-9 (B) and thrombus formation on HMEC-1 lines (C) at baseline, at 1–4 days and 7–16 days of CPAP ventilator support and at post discharge recovery visit in eculizumab-treated patients (Black columns) and "biochemical" controls (diagonal stripes), respectively. Numbers under the columns describe the number of eculizumab—treated patients or controls evaluated at each time-point. Horizontal dashed lines are the upper and lower limit of normal range (mean±2SD) of each considered parameter.

Abbreviations: CPAP, Continuous Positive Airway Pressure; Ecul, Eculizumab. Data are mean ± SD. One-way Anova test., *P<0.05, **P<0.01, ***P<0.001 versus baseline; ^^P<0.01 versus recovery; ###P<0.0001 versus "biochemical controls" at the same time-point.

follow-up changes in both parameters significantly differed between groups during the first two weeks of CPAP ventilator support. At recovery visits all considered parameters were in normal range and similar between groups.

**Long-term endpoints.** Over a median (IQR) observation period of 47.0 (14.0–121.0) days, four of the ten eculizumab-treated patients died or were discharged with chronic complications as compared to 52 of the 65 controls (80.0%) (Fig 4A). Event rate was significantly lower in eculizumab-treated patients than in controls [$HR_{Crude}$ (95% CI): 0.26 (0.09–0.72), p = 0.010]. The difference between groups was statistically significant even after adjustment for age and sex [$HR_{Adjusted}$ (95% CI): 0.29 (0.10–0.84), p = 0.022], and further adjustment for baseline serum creatinine [$HR_{Adjusted}$ (95% CI): 0.30 (0.10–0.84), p = 0.023]. Causes of death and chronic complications at discharge are shown in S1 and S2 Tables, respectively.

Two eculizumab-treated patients versus 31 (47.7%) controls died. Mortality rate, however, did not significantly differ between groups, even after adjustment for age and sex, and further adjustment for baseline serum creatinine (Fig 4B). Six patients and 13 controls (20.0%) were discharged without chronic complications. The event rate was significantly higher in eculizumab-treated patients than in controls [$HR_{Crude}$ (95% CI): 2.88 (1.08–7.70), p = 0.035, (Fig 4C)]. Between-group difference approached the nominal significance after adjustment for age and sex [$HR_{Adjusted}$ (95% CI): 2.92 (0.99–8.67), p = 0.053], but was not significant after adjustment for baseline serum creatinine [$HR_{Adjusted}$ (95% CI): 2.21 (0.71–6.88), p = 0.171].

**Other endpoints.** During the observation period, 5 (50%) eculizumab-treated patients and 21 (32.3%) controls required mechanical ventilation (p = 0.301). There were six (9.2%) cardiovascular events (two cardiogenic shocks, one myocardial infarction, one atrial fibrillation, one atrioventricular block and one supraventricular tachycardia) and six (9.2%) thromboembolic events in controls versus none in eculizumab-treated patients. None of the patients in either group required renal replacement therapy.

**Correlation analyses.** At baseline, plasma sC5b-9 levels positively correlated with D-dimer concentrations ($r_s$ = 0.849, p = 0.002) and serum-induced C5b-9 deposits with respiratory rate ($r_s$ = 0.590, p = 0.026). At one-week follow-up, changes in plasma sC5b-9 levels versus baseline directly correlated with concomitant changes in D-dimer concentrations ($r_s$ = 0.925, p = 0.001). Changes in serum-induced C5b-9 deposits at one ($r_s$ = 0.706, p = 0.010, Fig 3E) and two ($r_s$ = 0.751, p = 0.032, Fig 3F) weeks positively correlated with concomitant changes in respiratory rate. At baseline, neutrophil-to-lymphocyte ratio (r = 0.675, p = 0.046) and platelet-to-lymphocyte ratio (r = 0.807, p = 0.009) directly correlated with thrombus formation. Changes in neutrophil-to-lymphocyte (r = 0.884, p = 0.046) and platelet-to-lymphocyte (r = 0.908, p = 0.033) ratios at one week of follow-up positively correlated with concomitant changes in thrombus formation.

## Discussion

In this fully academic, single centre, two-phase study we first found that eculizumab compassionate therapy was safe and well tolerated in ten patients with severe COVID-19 requiring CPAP ventilator support. No treatment-related adverse event was reported. Respiratory distress promptly improved, as documented by significant reduction in respiratory rate and concomitant increase in $PaCO_2$ at one and two weeks of CPAP ventilator support. Inflammation

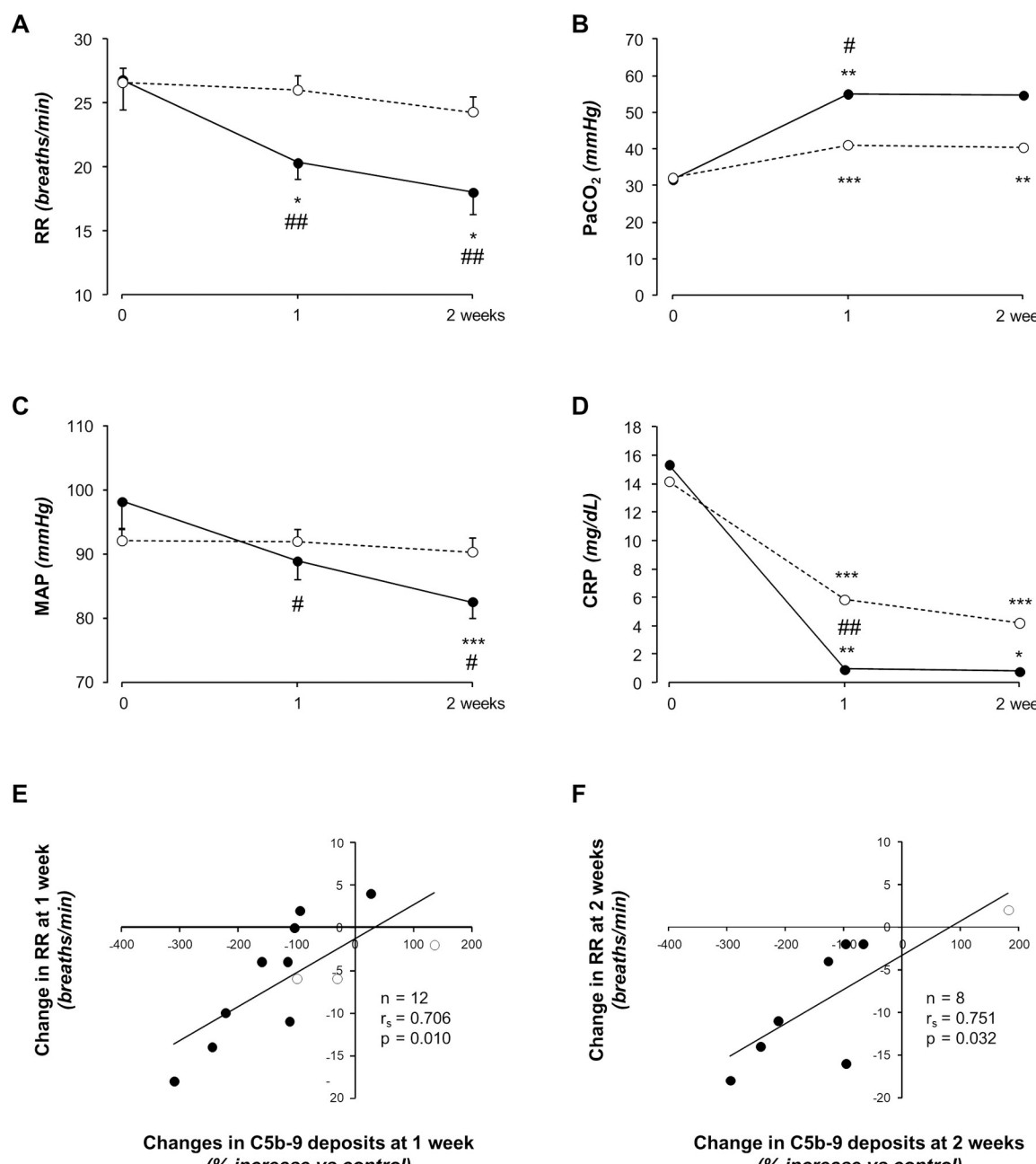

**Fig 3. Changes in clinical, respiratory and laboratory parameters in eculizumab-treated patients and all controls, and correlations between changes in respiratory rate and ex vivo C5b-9 deposition at different time points in eculizumab-treated patients and "biochemical" controls considered as a whole.** Respiratory rate (A), $PaCO_2$ (B), mean arterial pressure (C) and C-reactive protein plasma levels (D) at baseline and at 1–4 days and 7–16 days of CPAP ventilator support in eculizumab-treated patients (black circles and black continuous line) and in controls (white circles and grey dashed line). Data are mean ± SEM or median. Correlations between changes in serum-induced C5b-9 deposits at 1 week (E) and 2 weeks (F) of CPAP ventilator support and concomitant changes in respiratory rate in eculizumab-treated patients (black circles) and controls (white circles) considered as a whole. Number of patients, $r_s$ and p values for each correlation are shown in the two panels. Abbreviations: CRP, C-reactive protein; MAP, mean arterial pressure; RR, respiratory rate. *$P<0.05$, **$P<0.01$, ***$P<0.001$ versus baseline. #$P<0.05$, ##$P<0.01$ versus controls.

A

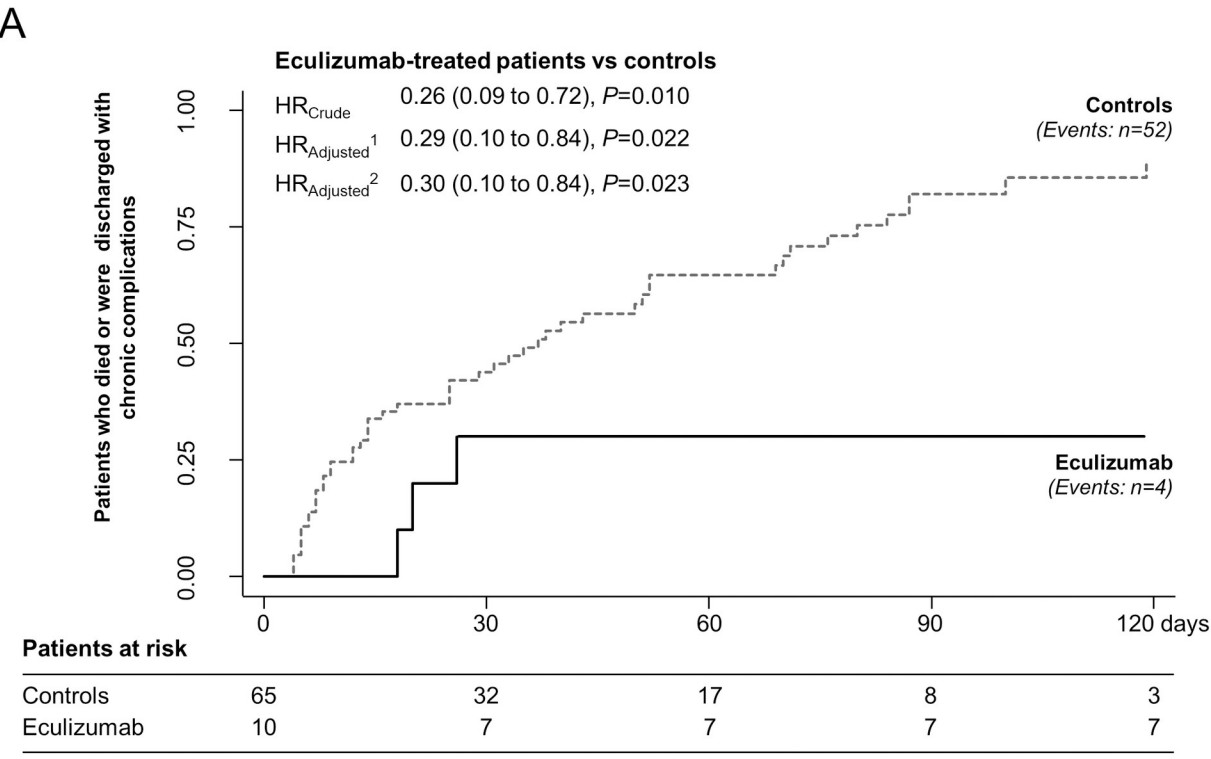

B
C

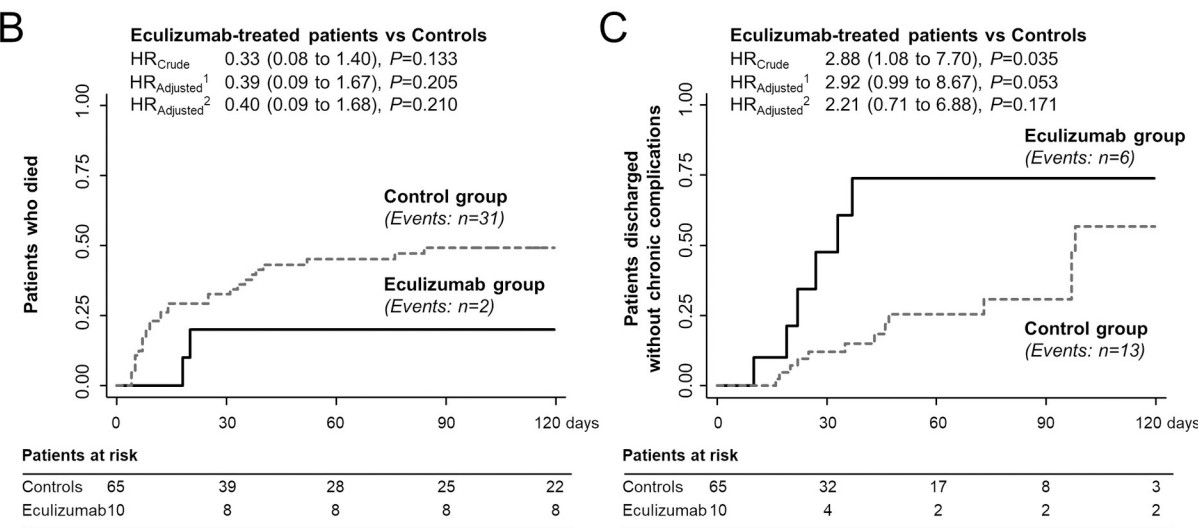

**Fig 4. Kaplan-Meier curves for the combined endpoint of mortality or discharge with chronic complications, and for mortality as a single endpoint and discharge without chronic complications in eculizumab-treated patients and controls.** Kaplan-Meier curves show the proportion of eculizumab-treated patients and controls who reached the combined endpoint of mortality or discharge with disabling chronic complications (A), mortality as a single endpoint (B) or discharge without chronic complications (C) during the observation period. Hazard ratios (HRs) and 95% confidence intervals are crude and adjusted for sex and age classes (model 1), or for sex, age classes and baseline serum creatinine (model 2). The number of patients at risk is shown in the bottom table. Black continuous line, eculizumab group; grey dashed line, control group.

also improved as demonstrated by reduction in CRP levels at the same time points. Conversely, in the second controlled phase we found that in controls respiratory rate, $PaCO_2$ and CRP levels did not change appreciably at one and two weeks of follow-up. Finding that at one and two weeks changes in these parameters significantly differed between groups confirmed that eculizumab remarkably and promptly improved respiratory distress and inflammation during the acute phase of the disease versus standard therapy alone.

Over a median observation period of 47 days, eculizumab-treated patients were also significantly protected against the combined endpoint of in-hospital death or discharge with invalidating chronic sequelae, including residual respiratory insufficiency in most cases. Moreover eculizumab associated with increased probability of discharge without chronic complications and a trend to better survival rate versus controls. Treatment effect on the combined endpoint was significant even after adjustment for age, sex and baseline serum creatinine. On the other hand, treatment effect on mortality rate considered as a single endpoint failed to reach the nominal significance most likely because of the too small number of events. Eculizumab therapy also reduced arterial blood pressure versus controls. Whether this effect was explained by reduced sympathetic tone associated with amelioration of respiratory distress and/or by blunted complement-mediated arteriolar vasoconstriction remains elusive [26].

Study findings were unlikely confounded by differences in the distribution of risk factors or in patient's care because at inclusion patients' characteristics were quite similar between groups, and all patients were followed by the same "COVID teams" during the same observation period and were managed according to the same standardised monitoring and treatment protocols. Findings were not confounded by concomitant experimental treatments because patients included in clinical trials were not considered. The more frequent use of RAS blockers in controls should not have confounded the results because these medications do not appear to affect disease progression and case-fatality in COVID-19 [27].

The sub-study in the ten eculizumab-treated patients and four "biochemical" controls showed a marked increase in plasma sC5b-9 levels along with extremely activated *ex vivo* complement and thrombus deposition on human cultured endothelial cells induced by patients' sera. SC5b-9 plasma levels decreased and *ex-vivo* C5b-9 and thrombus deposition fully normalised with eculizumab, and all parameters in eculizumab-treated patients normalised at recovery visits. In controls, plasma sC5b-9 levels and *ex vivo* C5b-9 and thrombus deposition did not appreciably change at one to four and seven to 16 days of follow-up versus baseline and normalised only at recovery visit. Between-group differences in *ex vivo* C5b-9 formation at the same time points and in thrombus deposition at one to four days were highly significant. Thus, eculizumab fully blunted *ex vivo* complement deposition and thrombogenesis induced by patients' SARS-CoV-2 infected sera. These data suggest that in COVID-19, disease severity could be sustained by extreme activation of the complement terminal pathway, both in the circulation and on the endothelial cell surface [19,20]. Benefits of C5 blockade could be mediated by prompt and effective systemic complement inhibition and protection from complement deposition and thrombus formation on endothelial cell surface during the acute phase of the disease. Consistently, respiratory rate positively correlated with *ex vivo* C5b-9 formation on endothelial cells at inclusion and its reduction at one to four days and seven to 16 days of follow up correlated with concomitant reductions in *ex vivo* C5b-9 formation.

In healthy subjects, complement modulates pro- and anti-inflammatory functions and facilitates the clearance of pathogens and apoptotic cells [28]. This modulatory function is disrupted by SARS-CoV-2 that triggers uncontrolled cleavage of the terminal complement protein C5 with consequent excess production of the proinflammatory anaphylatoxin C5a and of the terminal complement complex C5b-9. These changes activate endothelial and phagocytic cells and sustain production of reactive oxygen species [7]. The C5b-9 complex may also

directly injure endothelial and alveolar cells [29] with consequent disruption of the lung tissue. C5a and C5b-9 may also have pro-thrombotic effects by acting on endothelium, neutrophils and platelets [12,29,30]. Thus, C5 blockade by eculizumab might also serve to prevent the thromboembolic complications of COVID-19 [9]. These effects are achieved without affecting upstream immune-modulatory and immune-protective functions of the complement cascade [28], which might explain—along with antimicrobial prophylaxis against gram-positive encapsulated bacteria—why in our patients eculizumab treatment was not associated with excess risk of infectious complications. However, the risk of infection with Klebsiella pneumonie despite antimicrobial prophylaxis during C5-blocking therapy [31] should be taken into consideration because this gram-negative encapsulated bacterium often produces large spectrum beta-lactamases that can inactivate antibiotics commonly recommended to prevent meningococcal infection upon exposure to eculizumab [32]. The findings are not affected by doses of heparin or low molecular weight heparin that are used in clinics to prevent or treat pulmonary thromboembolism [21,33].

At variance with previous scanty reports [34], none of our patient disclosed signs of thrombotic microangiopathy. Lack of predisposing genetic abnormalities in the complement system could explain these findings and even suggests that SARS-CoV-2 infection is *per se* sufficient to induce complement activation and precipitate thrombo-embolic complications independent of host genetic predisposition [35].

Major study limitations were the relatively small number of patients receiving eculizumab therapy and the non-randomised and unblinded design. Major strengths were the controlled design and the integrated evaluation of potential mediators of the disease, markers of respiratory distress and long-term hard endpoints. All considered outcomes were pre-specified and data assessors were blinded to treatment. Treatment assignment was based on predefined guidelines, which limited the risk of investigator bias in the allocation to treatment groups. When we designed our study, few early doses of eculizumab had been reported to inhibit complement activation in severe cases of Shiga-toxin associated HUS [36]. With this background we administered only two 900 mg doses of eculizumab one week apart at acute onset of severe COVID-19. We found that this treatment protocol was safe and effective in our patients. Recent data suggest that intensified treatment compared to that indicated for atypical HUS appeared to improve outcomes of patients with COVID-19, but was associated with excess infectious complications in particular of life-threatening ventilator-associated pneumonia [15]. However, assessing whether higher eculizumab doses, shorter intervals between drug administrations or longer treatment periods would have been more effective, or rather less safe, was beyond our purposes. We did not measure the degree of circulating C5 blockade by eculizumab through CH50 or similar assays. However finding that ex vivo serum-induced C5b-9 deposition on HMEC-1 fully normalised in COVID-19 patients after eculizumab, would support an effective degree of C5 inhibition as we previously documented in patients with aHUS [20].

In conclusion, our findings–that need confirmation in a prospective randomised clinical trial—suggest that adding only two 900 mg doses of eculizumab to standard therapy in patients with severe COVID-19 who were receiving CPAP support for 24 hours or less, can safely improve respiratory dysfunction and decrease the combined endpoint of long-term mortality and chronic complications. These findings may have major implications, since effective complement C5 blockade restricted to the early acute phase of the disease could have better risk/benefit profiles than standard or intensified treatement protocols. Optimised cost/effectiveness could also facilitate patients' access to compassionate treatment with this expensive medication, particularly in resource-restricted settings.

## Supporting information

**S1 Fig. Experimental design of ex vivo serum-induced thrombus formation experiments on microvascular endothelial cells.** HMEC-1 were treated with ADP and exposed for 2 h in static conditions to 50% serum from COVID-19 patients, from aHUS patients studied in remission (aHUS positive controls) or from control serum pool diluted with test medium. Perfusion of heparinised whole blood from healthy subjects (added with mepacrine) was then performed in a thermostatic flow chamber in which one surface of the perfusion channel was a glass slide seeded with a monolayer of endothelial cells, at a constant flow rate of 1500 sec$^{-1}$ (60 dynes/cm$^2$). After 3 min, perfusion was stopped, and the slide with the endothelial cell monolayer was dehydrated and fixed. The values were expressed in pixels$^2$.
(TIF)

**S1 Table. Causes of death in the study group as a whole (overall) and in cases and controls considered separately.**
(DOCX)

**S2 Table. Age, gender and chronic complications at hospital discharge in cases and controls.**
(DOCX)

**S1 Appendix. Supplementary methods.**
(DOCX)

**S2 Appendix. Study protocol.**
(PDF)

**S3 Appendix. STROBE checklist.**
(DOCX)

## Acknowledgments

Stefano Rota and Diego Curtò helped in patient identification and monitoring, Francesco Peraro contributed to statistical analyses, Davide Martinetti helped in the finalisation of the database and data extraction, Matteo Breno performed bioinformatic analysis of results of next generation sequencing and Lucia Liguori analysed the SNPs for the CFH H3 haplotype, Prof. Andrea Remuzzi setted up the perfusion chamber. We thank all doctors and nurses of the COVID Units who managed the patients and helped in their identification and treatment according to protocol guidelines.

## Author Contributions

**Conceptualization:** Piero Ruggenenti, Fabiano Di Marco, Giuseppe Remuzzi.

**Data curation:** Piero Ruggenenti, Monica Cortinovis, Federica Tomatis, Silvia Bernardi, Olimpia Diadei.

**Formal analysis:** Annalisa Perna, Tobia Peracchi.

**Investigation:** Fabiano Di Marco, Luca Lorini, Silvia Sala, Luca Novelli, Federico Raimondi, Sara Gastoldi, Miriam Galbusera, Roberta Donadelli, Caterina Mele, Rossella Piras, Marina Noris, Valentina Portalupi, Laura Cappelletti, Camillo Carrara, Ariela Benigni.

**Supervision:** Piero Ruggenenti, Giuseppe Remuzzi.

**Writing – original draft:** Piero Ruggenenti, Giuseppe Remuzzi.

**Writing – review & editing:** Piero Ruggenenti, Fabiano Di Marco, Monica Cortinovis, Luca Lorini, Silvia Sala, Luca Novelli, Federico Raimondi, Sara Gastoldi, Miriam Galbusera, Roberta Donadelli, Caterina Mele, Rossella Piras, Marina Noris, Valentina Portalupi, Laura Cappelletti, Camillo Carrara, Federica Tomatis, Silvia Bernardi, Annalisa Perna, Tobia Peracchi, Olimpia Diadei, Ariela Benigni, Giuseppe Remuzzi.

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
