## [Decision Letter · Decision Letter 0]

7 Oct 2021

PONE-D-21-17700Eculizumab in patients with severe coronavirus disease 2019 (COVID-19) requiring continuous positive airway pressure ventilator support: retrospective cohort studyPLOS ONE

Dear Dr. Remuzzi,

Thank you for submitting your manuscript to PLOS ONE. After careful consideration, we feel that it has merit but does not fully meet PLOS ONE’s publication criteria as it currently stands. Therefore, we invite you to submit a revised version of the manuscript that addresses the points raised during the review process.

Both reviewers, as well as the academic editor had significant concerns. Specifically, the description of the study was judged to be confusing and the conclusions may not be fully supported by the data considering the limited number of patients studied. There are also a variety of other variables/outcomes that could be included in the description of the patient populations, as outlined by reviewer 2. Further, a bit more context, in terms of relationships to other studies (see Reviewer 1) and to an ongoing clinical trial (see Reviewer 2), is required.  

We look forward to receiving your revised manuscript.

Kind regards,

Ruud AW Veldhuizen

Academic Editor

PLOS ONE

Additional Editor Comments (if provided):

1) As a non-expert reader of his manuscript, I found the description of the study protocols and patient populations very confusing. It seems this is a retrospective study but written as a prospective study.

2) The authors conclude: "Eculizumab safely improved respiratory dysfunction and long-term outcomes of patients with severe COVID-19" (Abstract and a similar statement in the discussion). Considering this is a non-blinded study, with only 10 patients receiving the in the drug of interest, in an extremely complex disease, the authors should downplay this conclusion. This includes being specific regarding their findings and staying away from statements like "long-term outcomes" which can easily be misinterpreted by readers. After reading this manuscript, my conclusion would be that this data support the safety and rationalizes a more extensive, blinded, multi-center trial for the use of Eculizumab in this patient population, or something of that nature.

Journal Requirements:

2. Thank you for including your ethics statement: "The compassionate treatment protocol and the controlled study were both approved by the local Ethical Committee. Participants provided written informed consent. Local Ethical Committee approved the study. " 

4. Thank you for stating in your Funding Statement: " The ASST Papa Giovanni XXIII in Bergamo (Italy) sponsored the trial, Brembo SpA (Curno, Bergamo, Italy) partially covered study costs by a liberal grant under the initiative "Progetto TrexUno" and Alexion Pharma Italy S.R.L. (Milan, Italy) freely supplied the study drug. Neither the sponsor nor the companies had any role in study design; in the collection, analysis and interpretation of data; in the writing of the report; and in the decision to submit the paper for publication. All authors had full access to all the data in the study and accept responsibility to submit for publication. "

5. Thank you for stating the following financial disclosure: "The ASST Papa Giovanni XXIII in Bergamo (Italy) sponsored the trial, Brembo SpA (Curno, Bergamo, Italy) partially covered study costs by a liberal grant under the initiative "Progetto TrexUno" and Alexion Pharma Italy S.R.L. (Milan, Italy) freely supplied the study drug. Neither the sponsor nor the companies had any role in study design; in the collection, analysis and interpretation of data; in the writing of the report; and in the decision to submit the paper for publication. All authors had full access to all the data in the study and accept responsibility to submit for publication." 

We note that one or more of the authors is affiliated with the funding organization, indicating the funder may have had some role in the design, data collection, analysis or preparation of your manuscript for publication; in other words, the funder played an indirect role through the participation of the co-authors. If the funding organization did not play a role in the study design, data collection and analysis, decision to publish, or preparation of the manuscript and only provided financial support in the form of authors' salaries and/or research materials, please do the following:

a. Review your statements relating to the author contributions, and ensure you have specifically and accurately indicated the role(s) that these authors had in your study. These amendments should be made in the online form.

b. Confirm in your cover letter that you agree with the following statement, and we will change the online submission form on your behalf: 

“The funder provided support in the form of salaries for authors [insert relevant initials], but did not have any additional role in the study design, data collection and analysis, decision to publish, or preparation of the manuscript. The specific roles of these authors are articulated in the ‘author contributions’ section."

6. Thank you for stating the following in the Competing Interests section: "M.G. reported grants Omeros Corporation, Alexion Pharmaceutical, F. Hoffman-La Roche Ltd and Novartis Pharma AG (payments were made to her institution); M.N. reported grants from Omeros Corporation, Novartis Pharma AG, Roche and BioCryst Pharmaceutical (payments were made to her institution) as well as personal fees from Inception Sciences and BioCryst Pharmaceutical. A.B. reported personal fees from Akebia Pharmaceuticals, Alexion Pharmaceutical, BioCryst Pharmaceutical, Janssen Research & Development LLC, as well as speaker honorarium/travel reimbursements from Alnylam, Boehringer Ingelheim and Inception Science Canada. G.R. reported personal fees from Akebia Pharmaceuticals, Alexion Pharmaceutical, BioCryst Pharmaceutical and Janssen Research & Development LLC, as well as speaker honorarium/travel reimbursements from Alnylam, Boehringer Ingelheim and Inception Science Canada. All the other authors have nothing to disclose."

We note that you received funding from a commercial source: Omeros Corporation, Alexion Pharmaceutical, F. Hoffman-La Roche Ltd, Novartis Pharma AG, Novartis Pharma AG, Roche and BioCryst Pharmaceutical, Akebia Pharmaceuticals, and Janssen Research & Development LLC.

7. Your ethics statement should only appear in the Methods section of your manuscript. If your ethics statement is written in any section besides the Methods, please move it to the Methods section and delete it from any other section. Please ensure that your ethics statement is included in your manuscript, as the ethics statement entered into the online submission form will not be published alongside your manuscript. 

Reviewers' comments:

Reviewer's Responses to Questions

**Comments to the Author**

1. Is the manuscript technically sound, and do the data support the conclusions?

Reviewer #1: Yes

Reviewer #2: Partly

2. Has the statistical analysis been performed appropriately and rigorously? 

Reviewer #1: Yes

Reviewer #2: Yes

3. Have the authors made all data underlying the findings in their manuscript fully available?

Reviewer #1: Yes

Reviewer #2: Yes

4. Is the manuscript presented in an intelligible fashion and written in standard English?

Reviewer #1: Yes

Reviewer #2: Yes

5. Review Comments to the Author

Reviewer #1: This is a good, comprehensive retrospective cohort study using Eculizumab in patients with severe coronavirus disease 2019 (COVID-19). However, some omissions should be addressed.

1. Authors should explain why the standard dose of LMWH may be sufficient or insufficient, as in some reported cases (Stattin K, et al. Inadequate prophylactic effect of low-molecular weight heparin in critically ill COVID-19 patients. J Crit Care. 2020 Dec; 60:249-252). Taking into account that LMWH significantly suppresses the cleavage of C3 (Amara U, et al., Molecular intercommunication between the complement and coagulation systems. J Immunol. 2010 Nov 1;185(9):5628-36), and therefore the levels of sC5b-9.

2. The authors need to explain in Table 2, during the follow-up of the study group (Baseline, Week 1 and Week 2) with Eculizumab; why the levels of D-Dimer increase slightly, however, the thrombus formation decreases, at least during the week1?

3. In addition, Table 2 could include the neutrophils to lymphocytes ratio and the platelets to lymphocytes ratio in order to discuss and correlate them with laboratory parameters.

4. The authors mention in lines 322-324 “Mortality rate, however, did not significantly differ between groups, even after adjustment for age and sex, and further adjustment for baseline serum creatinine (Fig 4B)”; but they should explain this.

5. The authors should include the pH of the patients in Table 2. Taking into consideration that there could be some correlation with complement activation in patients with severe COVI-19 since a relationship has been shown in-vitro (Fishelson Z, Horstmann RD, Müller-Eberhard HJ. Regulation of the alternative pathway of complement by pH. J Immunol. 1987 May 15;138(10):3392-5).

6. Considering that complement and thrombosis converge in the author´s research, they should cite Bont CM, et al., NETosis, complement, and coagulation: a triangular relationship. Cell Mol Immunol. 2019; 16 (1): 19-27. doi: 10.1038 / s41423-018-0024-0.

Reviewer #2: In this paper, the authors describe the use of eculizumab at the onset of COVID-19 related ARDS as reflected by the need for CPAP. Eculizumab, a monoclonal antibody against complement protein C5, was used in 10 patients with COVID (each patient received two doses 7-10 days apart) and compared them to a cohort of COVID-19 patients who did not receive eculizumab, cared for by the same team. They measured plasma C5b-9, generated ‘in vitro’ C5b-9, and thrombus formation from patient samples, at various time points. The authors describe that the eculizumab group showed improvement in the respiratory rate (primary outcome) at 1 and 2 weeks compared to their baseline, while the control group did not have a similar change. Additionally, they noticed the eculizumab respiratory rate correlated with the ex vivo C5b-9 deposits. This report is clinically relevant in highlighting the role of complement in COVID-19 pathophysiology, especially since a targetable drug such as eculizumab is already being used for other diseases.

1. From a methodology perspective, the manuscript has been written confusingly about this being a prospective and a retrospective study (line 141). Additionally, the patients who received eculizumab have been reported to be ‘every 6th or 7th patient’ with COVID. At the same time, the manuscript also reports that it was the treating physician's discretion as to who gets eculizumab (lines 104 and 107). These discrepancies concern how this study was performed and how the data was analyzed. There should be a better justification to why no sample size calculation was performed while primary and secondary objectives were defined.

2. The described study and details of the NCT04288713 study on clinicaltrials.gov do not match. The latter is registered in the United States, and reports of eculizumab dosing every seven days, and the endpoints are also different. The authors should explain this discrepancy and why this deviation in the protocol was made.

3. The methods section should include the time period of this study, as there have been several studies looking at complement blockade in COVID-19. Some studies and case reports have demonstrated increased use of higher doses of eculizumab or more frequent dosing in COVID. These patients with COVID have been reported to have a higher level of complement activation than aHUS or PNH. Was complement blockade examined in the treated patients, along with other complement activation markers?

4. Can the authors comment on why the in vitro C5b-9 at baseline in the control group (Fig 2b) was higher than their plasma sC5b-9 values (as shown in figure 2A)?

5. The two groups have different levels of inflammation even though the p-value wasn’t significant. The group which received eculizumab had higher sC5b-9, higher PaO2, lower LDH, creatinine, GFR and D-dimer. This trend with the eculizumab group being slightly milder at the onset and reportedly had a better response to eculizumab. Even though there was a smaller ‘n’, it may be helpful to match and perform sub-analyses of the Ecu group with ‘selected patients’ from the non-Ecu group with similar respiratory and laboratory profiles.

6. The control group had elevated D-Dimer levels, and this high level has been shown to be associated with poor outcomes in several COVID-19 studies. Can the reported increase in ex vivo thrombus formation in the control group be attributed to this high baseline D-dimer with or without evidence of pulmonary embolus, contributing to the poorer respiratory outcome? The ‘n’ of 3 and 2 in the control group for thrombus formation assays is severely underpowered and questions the validity of the data.

7. Authors should report if patients in either group required mechanical ventilation, renal replacement therapy, cardiac dysfunction, systemic thrombosis, antiviral medications, among other outcomes. This would help ascertain if other endpoints were or were not different between these two groups.

8. Can authors also report which patients received hydroxychloroquine and remdesivir and how it may have contributed to their response to COVID-19 infection.

6. PLOS authors have the option to publish the peer review history of their article (what does this mean?). If published, this will include your full peer review and any attached files.

Reviewer #1: No

Reviewer #2: No

---

## [Author Response · Author response to Decision Letter 0]

27 Oct 2021

Additional Editor Comments:

1) As a non-expert reader of his manuscript, I found the description of the study protocols and patient populations very confusing. It seems this is a retrospective study but written as a prospective study.

We acknowledge that the study design was complex, but this was due to reasons that were largely beyond our control. As soon as we got the approval by the Ethical Committee we used all supplied medication to prospectively treat all approachable patients (from February 2020 to April 2020) in the context of the FDA program of eculizumab off-label compassionate use for the treatment of non-intubated patients with COVID-19. The compassionate use implied that patients could not be treated in the context of a controlled study. When the last eculizumab-treated patient had completed the treatment period (and therefore the phase of compassionate therapy had been completed) we identified patients who required CPAP ventilator support at the same institution and during the same period (from February 2020 to April 2020), and fulfilled the same eligibility criteria for eculizumab therapy but for practical reasons could not access to eculizumab (see below). These patients served as controls. Then, baseline and outcome data from eculizumab-treated patients and from non-eculizumab treated controls were recorded and analysed in the context of a retrospective controlled study. Thus, eculizumab patients were prospectively treated whereas data from eculizumab patients and non-eculizumab controls were retrospectively recorded in the context of a controlled study. We clarified this important issue in the revised version of the manuscript (Page 5, lines 94-96).

As for the description of the study populations it must be taken into consideration that the study was performed during the first, dramatic wave of the COVID-19 pandemic in Lombardy (Italy). In actual facts, during that emergency period it was virtually impossible to treat with eculizumab and adequately monitor all consecutive patients who required CPAP ventilator support. Thus, the selection of eculizumab recipients was at the discretion of the treating physician and influenced by logistic reasons. As clarified in the Methods of the manuscript, “to avoid overlaps of drug administrations and specific laboratory tests to monitor treatment effects in different patients, and prevent possible interference with clinical patient management, we administered eculizumab to one patient every six to seven consecutive potential candidates”. Then, we identified patients who required CPAP ventilator support during the same inclusion period but who could not access to eculizumab. We finally included into the study eculizumab treated patients and non-eculizumab treated controls. This is why there was no sample size calculation for this study. In the revised manuscript we better clarified this important issue (Page 5, lines 94-96).

2) The authors conclude: "Eculizumab safely improved respiratory dysfunction and long-term outcomes of patients with severe COVID-19" (Abstract and a similar statement in the discussion). Considering this is a non-blinded study, with only 10 patients receiving the in the drug of interest, in an extremely complex disease, the authors should downplay this conclusion. This includes being specific regarding their findings and staying away from statements like "long-term outcomes" which can easily be misinterpreted by readers. After reading this manuscript, my conclusion would be that this data support the safety and rationalizes a more extensive, blinded, multi-center trial for the use of Eculizumab in this patient population, or something of that nature.

We agree with the comment of the Reviewer that we have to tone down the emphasis on our conclusions because they are based on data that were obtained from only 10 treated patients and that were analysed in the context of a non-blinded study. As suggested in the conclusions of the abstract and discussion of the revised manuscript we emphasized that our findings need confirmation in a prospective randomized clinical trial. We also replaced “long-term outcome” with “combined endpoint of mortality and discharge with chronic complications” (Page 3, lines 57-59; Page 25, lines 461-465).

Journal Requirements:

We have amended our manuscript according to PLOS ONE's style requirements.

2. Thank you for including your ethics statement: "The compassionate treatment protocol and the controlled study were both approved by the local Ethical Committee. Participants provided written informed consent. Local Ethical Committee approved the study. " 

In the revised version of the manuscript, we have specified that the Ethical Committee of Bergamo approved the study (Page 5, lines 97-98).

We have amended the “Ethics Statement” field of the submission form as required.

We ensure to have provided the same grant information in the ‘Funding Information’ and ‘Financial Disclosure’ sections.

Please consider that Brembo SpA (Curno, Bergamo, Italy) partially covered study costs by a liberal grant. Thus, there was not a grant number.

4. Thank you for stating in your Funding Statement: " The ASST Papa Giovanni XXIII in Bergamo (Italy) sponsored the trial, Brembo SpA (Curno, Bergamo, Italy) partially covered study costs by a liberal grant under the initiative "Progetto TrexUno" and Alexion Pharma Italy S.R.L. (Milan, Italy) freely supplied the study drug. Neither the sponsor nor the companies had any role in study design; in the collection, analysis and interpretation of data; in the writing of the report; and in the decision to submit the paper for publication. All authors had full access to all the data in the study and accept responsibility to submit for publication. "

We have included the amendment Funding Statement within our cover letter.

5. Thank you for stating the following financial disclosure: "The ASST Papa Giovanni XXIII in Bergamo (Italy) sponsored the trial, Brembo SpA (Curno, Bergamo, Italy) partially covered study costs by a liberal grant under the initiative "Progetto TrexUno" and Alexion Pharma Italy S.R.L. (Milan, Italy) freely supplied the study drug. Neither the sponsor nor the companies had any role in study design; in the collection, analysis and interpretation of data; in the writing of the report; and in the decision to submit the paper for publication. All authors had full access to all the data in the study and accept responsibility to submit for publication." 

We note that one or more of the authors is affiliated with the funding organization, indicating the funder may have had some role in the design, data collection, analysis or preparation of your manuscript for publication; in other words, the funder played an indirect role through the participation of the co-authors. If the funding organization did not play a role in the study design, data collection and analysis, decision to publish, or preparation of the manuscript and only provided financial support in the form of authors' salaries and/or research materials, please do the following:

a. Review your statements relating to the author contributions, and ensure you have specifically and accurately indicated the role(s) that these authors had in your study. These amendments should be made in the online form.

We ensure to have specifically and accurately indicated the roles that the authors affiliated to the ASST Papa Giovanni XXIII in Bergamo (Italy) had in our study.

b. Confirm in your cover letter that you agree with the following statement, and we will change the online submission form on your behalf:

 “The funder provided support in the form of salaries for authors [insert relevant initials], but did not have any additional role in the study design, data collection and analysis, decision to publish, or preparation of the manuscript. The specific roles of these authors are articulated in the ‘author contributions’ section."

We confirm our agreement with the above sentence, that was included within the cover letter.

6. Thank you for stating the following in the Competing Interests section: "M.G. reported grants Omeros Corporation, Alexion Pharmaceutical, F. Hoffman-La Roche Ltd and Novartis Pharma AG (payments were made to her institution); M.N. reported grants from Omeros Corporation, Novartis Pharma AG, Roche and BioCryst Pharmaceutical (payments were made to her institution) as well as personal fees from Inception Sciences and BioCryst Pharmaceutical. A.B. reported personal fees from Akebia Pharmaceuticals, Alexion Pharmaceutical, BioCryst Pharmaceutical, Janssen Research & Development LLC, as well as speaker honorarium/travel reimbursements from Alnylam, Boehringer Ingelheim and Inception Science Canada. G.R. reported personal fees from Akebia Pharmaceuticals, Alexion Pharmaceutical, BioCryst Pharmaceutical and Janssen Research & Development LLC, as well as speaker honorarium/travel reimbursements from Alnylam, Boehringer Ingelheim and Inception Science Canada. All the other authors have nothing to disclose."

We note that you received funding from a commercial source: Omeros Corporation, Alexion Pharmaceutical, F. Hoffman-La Roche Ltd, Novartis Pharma AG, Novartis Pharma AG, Roche and BioCryst Pharmaceutical, Akebia Pharmaceuticals, and Janssen Research & Development LLC.

We have included the amended Competing Interests Statement within our cover letter.

7. Your ethics statement should only appear in the Methods section of your manuscript. If your ethics statement is written in any section besides the Methods, please move it to the Methods section and delete it from any other section. Please ensure that your ethics statement is included in your manuscript, as the ethics statement entered into the online submission form will not be published alongside your manuscript. 

As required, we have deleted the Ethics Statement from the Abstract section of the revised manuscript.

Reviewers' comments:

Reviewer's Responses to Questions

Comments to the Author

1. Is the manuscript technically sound, and do the data support the conclusions?

Reviewer #1: Yes

Reviewer #2: Partly

2. Has the statistical analysis been performed appropriately and rigorously?

Reviewer #1: Yes

Reviewer #2: Yes

3. Have the authors made all data underlying the findings in their manuscript fully available?

Reviewer #1: Yes

Reviewer #2: Yes

4. Is the manuscript presented in an intelligible fashion and written in standard English?

Reviewer #1: Yes

Reviewer #2: Yes

5. Review Comments to the Author

Reviewer #1: This is a good, comprehensive retrospective cohort study using Eculizumab in patients with severe coronavirus disease 2019 (COVID-19). However, some omissions should be addressed.

We thank the Reviewer for the rewarding comments on our work. Please find below all our point-by-point answers to the comments of the Reviewer.

1. Authors should explain why the standard dose of LMWH may be sufficient or insufficient, as in some reported cases (Stattin K, et al. Inadequate prophylactic effect of low-molecular weight heparin in critically ill COVID-19 patients. J Crit Care. 2020 Dec; 60:249-252). Taking into account that LMWH significantly suppresses the cleavage of C3 (Amara U, et al., Molecular intercommunication between the complement and coagulation systems. J Immunol. 2010 Nov 1;185(9):5628-36), and therefore the levels of sC5b-9.

The Reviewer raised an important point. Actually, there is a cross-link between coagulation and the complement system. Heparin and LMWH have been shown to inhibit complement activation in vitro, however this effect is observed only with supratherapeutic concentrations of the drug. For instance, in the J Immunol paper by Amara et al. 25 micrograms/ml fondaparinux were required to significantly reduce C5a generation in vitro in human serum, a concentration much higher than therapeutic Cmax levels of the drug (0.3-0.7 micrograms/ml). In addition, in a previously published study we found that addition of 10 U/ml heparin to serum from patients with atypical hemolytic uremic syndrome (aHUS) before incubation with HMEC-1 did not reduce the intensity of C5b-9 staining (Bettoni S et al J Immunol 2017; 199:1021-1040). 

Altogether, the above findings indicate that doses of LMWH given to COVID-19 patients are not enough to control the burst of complement activation caused by SARS-CoV-2 infection. We briefly addressed this important point in the Discussion of the revised manuscript (Page 24, lines 434-436).

2. The authors need to explain in Table 2, during the follow-up of the study group (Baseline, Week 1 and Week 2) with Eculizumab; why the levels of D-Dimer increase slightly, however, the thrombus formation decreases, at least during the week1?

D-dimer test reflects progressive reabsorption of preformed intravascular thrombi. The ex vivo test, on the contrary, assesses the formation of thrombi on the surface of HMEC-1 lines induced by patient sera. Thus the two tests describe two different phenomena. The D-Dimer test measures the consequence of an event that has already occurred or that is occurring. Thus the D-Dimer test increase follows or parallels thrombi formation. The ex vivo test measures an activity that is present in patient sera that induces (and therefore precedes) thrombus formation. Thus, it is expected that inhibition of serum thrombogenic activity (documented by the ex vivo test) precedes the inhibition of thrombi formation which in turn precedes the decrease in the D-Dimer test. Thus, normalisation in ex vivo thrombus formation is expected to precede D-Dimer test normalisation. This could explain why at week 2 ex vivo thrombus formation was inhibited whereas the D-Dimer test was still increased.

3. In addition, Table 2 could include the neutrophils to lymphocytes ratio and the platelets to lymphocytes ratio in order to discuss and correlate them with laboratory parameters.

As suggested by the Reviewer, in the revised version of the manuscript we included neutrophil-to-lymphocyte ratio and platelet-to-lymphocyte ratio at baseline (Table 1) and during follow-up (Table 2) according to study group. In the main text we specified that baseline characteristics were similar between eculizumab-treated patients and controls (Table 1), with the exception of lower platelet count and platelet-to-lymphocyte ratio, and more frequent use of renin-angiotensin system (RAS) blockers in controls (Page 17, lines 294-296).

At baseline, thrombus formation directly correlated with neutrophil-to-lymphocyte ratio (n=9, r=0.675, p=0.046) and with platelet-to-lymphocyte ratio (n=9, r=0.807, p=0.009). Changes in thrombus formation at one week follow-up positively correlated with changes in neutrophil-to-lymphocyte ratio (n=5, r=0.884, p=0.046) and platelet-to-lymphocyte (n=5, r=0.908, p=0.033) over the same time period. We thank the Reviewer for the suggestion to perform these analyses and briefly mentioned the results in the revised version of the manuscript (Pages 20-21, lines 368-372).

4. The authors mention in lines 322-324 “Mortality rate, however, did not significantly differ between groups, even after adjustment for age and sex, and further adjustment for baseline serum creatinine (Fig 4B)”; but they should explain this.

We speculate that eculizumab induced terminal complement inhibition translates into reduced inflammation and thrombogenesis and that both effects are expected to translate into reduced morbidity and mortality. However, in our present study the number of fatal events (that were numerically but remarkably less frequent in eculizumab treated patients than in non- eculizumab treated controls), was too small to provide the statistical analyses with adequate power to detect a significant effect on patient mortality considered as a single end point. We briefly discussed this important issue in the revised version of the manuscript (Page 22, lines 392-394). 

5. The authors should include the pH of the patients in Table 2. Taking into consideration that there could be some correlation with complement activation in patients with severe COVI-19 since a relationship has been shown in-vitro (Fishelson Z, Horstmann RD, Müller-Eberhard HJ. Regulation of the alternative pathway of complement by pH. J Immunol. 1987 May 15;138(10):3392-5).

As suggested by the Reviewer, in the revised version of the manuscript we included arterial pH of the patients at baseline (Table 1) and during follow-up (Table 2) according to study group. At baseline, arterial pH did not correlate with plasma sC5b-9 levels, serum-induced C5b-9 deposits or thrombus formation. Similarly, changes in arterial pH at one week and two weeks of follow-up did not correlate with concomitant changes in plasma sC5b-9 levels, serum-induced C5b-9 deposits or thrombus formation.

6. Considering that complement and thrombosis converge in the author´s research, they should cite Bont CM, et al., NETosis, complement, and coagulation: a triangular relationship. Cell Mol Immunol. 2019; 16 (1): 19-27. doi: 10.1038 / s41423-018-0024-0.

As properly suggested by the Reviewer we quoted the mentioned reference in the revised version of the manuscript (Page 4, line 430).

Reviewer #2: In this paper, the authors describe the use of eculizumab at the onset of COVID-19 related ARDS as reflected by the need for CPAP. Eculizumab, a monoclonal antibody against complement protein C5, was used in 10 patients with COVID (each patient received two doses 7-10 days apart) and compared them to a cohort of COVID-19 patients who did not receive eculizumab, cared for by the same team. They measured plasma C5b-9, generated ‘in vitro’ C5b-9, and thrombus formation from patient samples, at various time points. The authors describe that the eculizumab group showed improvement in the respiratory rate (primary outcome) at 1 and 2 weeks compared to their baseline, while the control group did not have a similar change. Additionally, they noticed the eculizumab respiratory rate correlated with the ex vivo C5b-9 deposits. This report is clinically relevant in highlighting the role of complement in COVID-19 pathophysiology, especially since a targetable drug such as eculizumab is already being used for other diseases.

We thank the Reviewer for the rewarding comments on our work.

1. From a methodology perspective, the manuscript has been written confusingly about this being a prospective and a retrospective study (line 141). Additionally, the patients who received eculizumab have been reported to be ‘every 6th or 7th patient’ with COVID. At the same time, the manuscript also reports that it was the treating physician's discretion as to who gets eculizumab (lines 104 and 107). These discrepancies concern how this study was performed and how the data was analyzed. There should be a better justification to why no sample size calculation was performed while primary and secondary objectives were defined.

We were authorized by our Ethical Committee to administer eculizumab to our patients with severe COVID-19 in the context of the FDA program of eculizumab off-label compassionate use for the treatment of non-intubated patients with COVID-19. The drug was freely supplied by the manufacturer (Alexion Pharma Italy S.R.L., Milan). As soon as we got the approval by the Ethical Committee we used all supplied medication to prospectively treat all approachable patients from February 2020 to April 2020, that is during the first, dramatic wave of the COVID-19 pandemic. Because of the compassionate use of eculizumab these patients could not be included in a prospective controlled trial. We aimed to treat all patients in need. In actual facts, during that emergency period it was virtually impossible to treat and adequately monitor all consecutive patients who required CPAP ventilator support. Thus, the selection of eculizumab recipients was at the discretion of the treating physician and influenced by logistic reasons. Indeed, as clarified in the Methods of the manuscript, “to avoid overlaps of drug administrations and specific laboratory tests to monitor treatment effects in different patients, and prevent possible interference with clinical patient management, we administered eculizumab to one patient every six to seven consecutive potential candidates”. Then, we identified patients who required CPAP ventilator support during the same inclusion period but who could not access to eculizumab (for the reasons described above). Thus we included into the study eculizumab treated patients and non-eculizumab treated controls. This is why there was no sample size calculation for this study. In the revised manuscript we better clarified this important issue. We pointed out that treatment with eculizumab was prospective but that the controlled study that recorded and analysed data from eculizumab-treated patients and non-eculizumab controls was retrospective in nature (Page 5, lines 94-98). 

2. The described study and details of the NCT04288713 study on clinicaltrials.gov do not match. The latter is registered in the United States, and reports of eculizumab dosing every seven days, and the endpoints are also different. The authors should explain this discrepancy and why this deviation in the protocol was made.

The described study and details of the NCT04288713 study on clinicaltrials.gov do not match with our study because the NCT04288713 registration number refers to another study by another group.

3. The methods section should include the time period of this study, as there have been several studies looking at complement blockade in COVID-19. Some studies and case reports have demonstrated increased use of higher doses of eculizumab or more frequent dosing in COVID. These patients with COVID have been reported to have a higher level of complement activation than aHUS or PNH. Was complement blockade examined in the treated patients, along with other complement activation markers?

As reported in the original version of the manuscript, study participants were included from February 2020 to April 2020 (Page 2, line 36; Page 10, line 170).

The Reviewer is right. In a non-randomized proof-of-concept study (Annane D et al. EClinicalMedicine 2020; 28:100590) patients with severe COVID-19 and admitted to ICU were treated with standard care or with standard care plus eculizumab. The initial regimen in the first 10 patients consisted of 900 mg of eculizumab at day 1, 8, 15 and 22 of ICU admission. This dosage resulted in transient and incomplete inhibition of the terminal pathway, which led to protocol amendment and the subsequent patients received higher and more frequent doses of the drug. 

In our study, we did not measure the degree of circulating C5 blockade by eculizumab through CH50 or similar assays. However, finding that ex vivo serum-induced C5b-9 deposition on HMEC-1 fully normalized in COVID-19 patients after eculizumab, would support an effective degree of C5 inhibition as we previously documented in patients with aHUS (Galbusera M et al. Am J Kidney Dis 2019; 74:56-72) We added this piece of information in the Limitations paragraph of the revised version of the manuscript (Page 25, lines 456-459).

4. Can the authors comment on why the in vitro C5b-9 at baseline in the control group (Fig 2b) was higher than their plasma sC5b-9 values (as shown in figure 2A)?

The two assays have different meanings. Plasma sC5b-9 values reflect the activation of terminal complement pathway in fluid phase. Soluble C5b-9 formed in the circulation is complexed with Vitronectin (S Protein) and fails to insert into membranes. At variance, the test of serum-induced C5b-9 formation on HMEC-1 is an ex vivo index of terminal complement activation on cell surface with the insertion of lytic membrane attack complex. Thus, values of the two parameters cannot be compared.

5. The two groups have different levels of inflammation even though the p-value wasn’t significant. The group which received eculizumab had higher sC5b-9, higher PaO2, lower LDH, creatinine, GFR and D-dimer. This trend with the eculizumab group being slightly milder at the onset and reportedly had a better response to eculizumab. Even though there was a smaller ‘n’, it may be helpful to match and perform sub-analyses of the Ecu group with ‘selected patients’ from the non-Ecu group with similar respiratory and laboratory profiles.

We wish to point out that none of the differences mentioned by the Reviewer was statistically significant. However, we acknowledge that the lack of significance could be explained by the relatively small number of patients and it cannot be definitely ruled out the possibility that some of the mentioned differences might have some biological relevance, even if statistically non-significant. Thus, to address the suggestion of the Reviewer we performed sensitivity analyses by matching cases and controls (1:3) by all the parameters mentioned by the Reviewer with the exception of sC5b9 that was available in only 3 controls. The results of these additional analyses were consistent with the original results of the study. In particular, four of the ten eculizumab-treated patients died or were discharged with chronic complications as compared to 24 of the 30 matched controls (80%). As previously reported in the whole study population, the event rate was significantly lower in eculizumab-treated patients than in the 30 controls [HR (95CI): 0.28 (0.11-0.70), p=0.006]. Two of the eculizumab-treated patients vs 14 controls (47%) died. Consistently with data in the whole study group, mortality rate considered as a single endpoint did not significantly differ between groups [HR (95CI): 0.33 (0.08-1.28), p=0.109]. Finally, six eculizumab treated patients and six controls (20%) were discharged without chronic complications. Again, consistently with results of original analyses, the event rate was higher in eculizumab-treated patients than in controls [HR (95CI). 3.07 (0.99-9.54), p=0.053], even if the difference was only borderline significant because of the reduced sample size. We thank the Reviewer for the suggestion to perform these additional analyses that provided additional evidence of the robustness of our findings. 

6. The control group had elevated D-Dimer levels, and this high level has been shown to be associated with poor outcomes in several COVID-19 studies. Can the reported increase in ex vivo thrombus formation in the control group be attributed to this high baseline D-dimer with or without evidence of pulmonary embolus, contributing to the poorer respiratory outcome? The ‘n’ of 3 and 2 in the control group for thrombus formation assays is severely underpowered and questions the validity of the data.

Baseline D-dimer levels did not differ in the eculizumab and the control group (Table 2 baseline), but thereafter increased at week 1 and week 2 only in the control group, likely reflecting worsening of the disease as compared with the eculizumab group.

Very unlikely D-dimer levels would have influenced significantly the results of ex vivo thrombus formation assay, since patients’ serum was washed out from the HMEC-1 monolayer after the preincubation step, then thrombus formation was assessed by flowing the cells with whole blood from healthy volunteers.

We recognize that the numerosity in the control group is low due to limited quantity of blood that could be taken from so sick patients. However, data were very comparably elevated in the 3 patients analyzed at baseline and at 1 week, and were normal in both patients analyzed at recovery. 

7. Authors should report if patients in either group required mechanical ventilation, renal replacement therapy, cardiac dysfunction, systemic thrombosis, antiviral medications, among other outcomes. This would help ascertain if other endpoints were or were not different between these two groups.

In the revised version of Table 1 we have reported that 7 (70%) eculizumab-treated patients and 32 (49%) controls received antiviral therapy (darunavir and cobicistat combination).

During the observation period, 5 (50%) eculizumab-treated patients and 21 (32.3%) controls required mechanical ventilation (p=0.301). There were six (9.2%) cardiovascular events (two cardiogenic shocks, one myocardial infarction, one atrial fibrillation, one atrioventricular block and one supraventricular tachycardia) and six (9.2%) thromboembolic events in controls versus none in eculizumab-treated patients. No patient in either group required renal replacement therapy. We reported these endpoints in the “Other endpoints” paragraph that we have added in the revised version of the manuscript (Pages 19-20, lines 342-348).

8. Can authors also report which patients received hydroxychloroquine and remdesivir and how it may have contributed to their response to COVID-19 infection.

In the revised version of Table 1 we have reported that 9 (90%) eculizumab-treated patients and 42 (65%) controls received hydroxychloroquine (p=0.154). In the original version of the manuscript, we have accidentally indicated that most patients received remdesivir. Instead, none of the eculizumab-treated or control patients was given remdesivir, since during the first wave of COVID-19 the combination of darunavir and cobicistat was considered as first-line antiviral therapy at our institution. We thank the Reviewer for bringing to our attention this issue, that we have addressed in the revised version of the manuscript (Page 10, lines 180-181) and Table 1. In actual fact, 7 (70%) eculizumab-treated patients and 32 (49%) controls were given antiviral therapy (darunvir and cobicistat combination) (p=0.313). 

Study findings were unlikely confounded by concomitant treatment with hydroxychloroquine or the combination of darunvir and cobicistat since the proportion of patients given these medications did not differ appreciably between the two study groups. Moreover, available data do not support the use of hydroxychloroquine or the combination of darunvir and cobicistat for the treatment of hospitalized patients with COVID-19 (JAMA 2020; 324:2165-2176; Front Med (Lausanne) 2021; 8:639970).

---

## [Decision Letter · Decision Letter 1]

25 Nov 2021

Eculizumab in patients with severe coronavirus disease 2019 (COVID-19) requiring continuous positive airway pressure ventilator support: retrospective cohort study

PONE-D-21-17700R1

Dear Dr. Remuzzi,

We’re pleased to inform you that your manuscript has been judged scientifically suitable for publication and will be formally accepted for publication once it meets all outstanding technical requirements.

Kind regards,

Ruud AW Veldhuizen

Academic Editor

PLOS ONE

Additional Editor Comments (optional):

Reviewers' comments:

Reviewer's Responses to Questions

**Comments to the Author**

1. If the authors have adequately addressed your comments raised in a previous round of review and you feel that this manuscript is now acceptable for publication, you may indicate that here to bypass the “Comments to the Author” section, enter your conflict of interest statement in the “Confidential to Editor” section, and submit your "Accept" recommendation.

Reviewer #1: All comments have been addressed

Reviewer #2: All comments have been addressed

2. Is the manuscript technically sound, and do the data support the conclusions?

Reviewer #1: Yes

Reviewer #2: Yes

3. Has the statistical analysis been performed appropriately and rigorously? 

Reviewer #1: Yes

Reviewer #2: Yes

4. Have the authors made all data underlying the findings in their manuscript fully available?

Reviewer #1: Yes

Reviewer #2: Yes

5. Is the manuscript presented in an intelligible fashion and written in standard English?

Reviewer #1: Yes

Reviewer #2: Yes

6. Review Comments to the Author

Reviewer #1: The manuscript is technically understandable, well written, and the data support the conclusions, it is recommended to accept.

Reviewer #2: The authors have satisfactorily addressed the reviewer's comments and this have strengthened the quality of this paper.

7. PLOS authors have the option to publish the peer review history of their article (what does this mean?). If published, this will include your full peer review and any attached files.

Reviewer #1: No

Reviewer #2: No

---

## [Editor Report · Acceptance letter]

9 Dec 2021

PONE-D-21-17700R1 

Eculizumab in patients with severe coronavirus disease 2019 (COVID-19) requiring continuous positive airway pressure ventilator support: *retrospective cohort study*

Dear Dr. Remuzzi:

I'm pleased to inform you that your manuscript has been deemed suitable for publication in PLOS ONE. Congratulations! Your manuscript is now with our production department. 

Kind regards, 

on behalf of

Dr. Ruud AW Veldhuizen 

Academic Editor

PLOS ONE